# Hierarchical Parametrization with Gaussian Process for Bayesian Meta-Learning

## Abstract

Meta-learning has emerged as a key approach to preparing neural networks for deployment with limited training data. Mainstream solutions focus on parameter initialization across training episodes to enable rapid generalization, which can be interpreted as adjusting the episode-specific posterior based on a cross-episode prior distribution in the context of Bayesian meta-learning. Despite the demonstrated efficacy of this probabilistic meta-learning paradigm, existing methodologies encounter performance bottlenecks, particularly as the scale and number of episodes increase. A promising strategy involves the integration of a hyper-network, which establishes a parameter memorization space across diverse episodes. In this paper, we propose Hierarchical Parametrization with Gaussian Process (HP-GP), a novel probabilistic meta-learning method that leverages the power of Gaussian Process. By implementing the amortization network layer-wise with decoupling variational Gaussian Process and normalizing flow, HP-GP offers probabilistic parametrization for meta-learning while requiring minimal modifications to the network architecture. This enables flexible and scalable integration of meta-learning into existing neural networks. Our experiments demonstrate the flexibility and robust generalization of HP-GP, outperforming other popular meta-learning methods.

## 1 Introduction

Deep learning has revolutionized modern computational paradigms by shifting to a data-centric learning approach rather than relying on explicit logical inference (Dargan et al., 2020; Tagg, 2003). Despite its success, the heavy dependence on large datasets significantly hinders its applicability in real-world scenarios, triggering various attempts to further increase data utilization.

Intending to teach the model learning to learn, meta-learning seeks to enhance the efficiency and adaptability of learning algorithms on new episodes by training a meta-learner across a distribution of episodes, enabling rapid adaptation (Vilalta & Drissi, 2002; Hospedales et al., 2021). The knowledge shared across training episodes equips models to efficiently utilize small support sets and outperform conventional training paradigms on corresponding query sets. However, the exploitation of limited support sets inherently introduces uncertainty into the meta-learning framework (Nguyen et al., 2020). To address this, Bayesian hierarchical models (Yoon et al., 2018; Zhang et al., 2021a; Ravi & Beatson, 2019) have been proposed for uncertainty quantification in meta-learning. From a network architecture perspective, weight parametrization using auxiliary networks effectively represents a Bayesian realization of hyper-networks (Krueger et al., 2018; Ha et al., 2016). While research on probabilistic amortization networks for specific architectures continues to gain traction, there is a notable lack of model-agnostic probabilistic hyper-networks tailored to meta-learning.

Building upon these observations, we extend the hierarchical variational inference framework for meta-learning, as introduced in Ravi & Beatson (2019); Iakovleva et al. (2020); Gordon et al. (2019). Specifically, we propose HP-GP, a model-agnostic and episode-independent framework that employs Gaussian Processes (GP) with normalizing flow conversion as hyper-networks. Our approach utilizes a variational GP with input-output tensor decoupling to model intermediate variables with reduced computation cost, and a Inverse Auto-regressive Flow (IAF) (Kingma et al., 2016) for distribution adaptation, in an attempt to convert Gaussian posteriors into actual distributions in BNN parameterization (Wenzel et al., 2020; Izmailov et al., 2021). This design takes into account both flexibility and scalability, making HP-GP suitable for a wide range of meta-learning applications.

We evaluate HP-GP against state-of-the-art meta-learning algorithms in scenarios involving domain generalization and episode adaptation, with identical base model architectures. Across all evaluated settings, HP-GP demonstrates competitive performance, establishing itself as a versatile and lightweight solution within the probabilistic meta-learning paradigm. Notably, on the challenging non-mutually exclusive pose prediction task—a task designed to corrupt models' generalization capabilities by encouraging memorization of canonical orientations from meta-training objects—HP-GP not only achieves superior performance metrics but also exhibits robustness to potential mismatches between query and support sets. This robustness is achieved through the exploitation of the combination of feed-forward parameterization for individual inputs and an expanded cross-class latent space modeling facilitated by the expressive probabilistic hyper-network. According to the experimental outcomes and theoretical guarantees, HP-GP offers a flexible, efficient, and robust framework that excels in both domain generalization and task adaptation scenarios, even on the most challenging meta-learning tasks.

## 2 RELATED WORK

Bayesian meta-learning, a subfield of meta-learning, seeks to model prior knowledge transferring across episodes within the probabilistic framework. A common approach for designing Bayesian meta-learning methods involves extending Model-Agnostic Meta-Learning (MAML) (Finn et al., 2017) to a Bayesian framework (Jerfel et al., 2019). Lightweight Laplace Approximation for Meta-Adaptation (LLAMA) (Grant et al., 2018) optimizes episode-specific parameters via a maximum a posteriori (MAP) estimator, recasting MAML as a hierarchical Bayesian model. Bayesian Meta-Learning with Chaser Loss (BMAML) (Yoon et al., 2018) employs Stein Variational Gradient Descent for optimization, successfully integrating nonparametric variational inference into the gradient-based meta-learning framework.

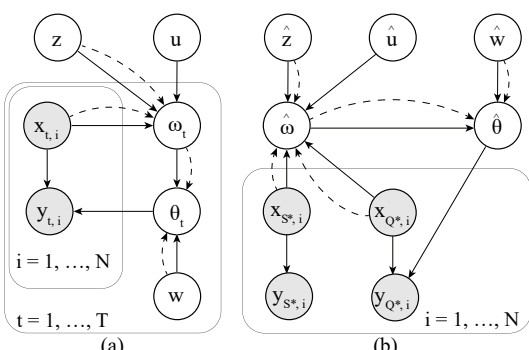

Figure 1: Graphical model of the proposed probabilistic meta-learning framework. Dotted lines represent variational approximations, and observed variables are shaded in gray. (a) HP-GP for meta-training scheme. (b) HP-GP for meta-testing scheme, with support and query set splitting. Detailed explanations can be found in Section 4.1.

Other methods explore the application of Bayesian learning models to the meta-learning paradigm (Kim & Hospedales, 2023; Jaques, 2018; Gordon et al., 2019). For instance, ALPaCA (Harrison et al., 2018) learns Bayesian regression by imposing a deep kernel on a Bayesian linear regression model, with the basis functions implemented by deep neural networks. PAC-Bayesian theory (Rothfuss et al., 2020) has also been leveraged to adapt Bayesian neural networks to novel episodes by crafting meta-learning priors.

The development of deep learning introduces hyper-network, a class of networks designed to parameterize base model, as a novel solution to enhancing network adaptability to dynamic episodes or environments (Ha et al., 2016; Krueger et al., 2018). However, the application is often limited to specific architectures or training procedures. For instance, VERSA (Gordon et al., 2019) replaces test-time optimization with forward passes through inference networks. Similarly, Karaletsos et al. (2018) employs hyper-networks to provide distributions for weights connecting units across layers. Other works, such as Goyal et al. (2020), focus on improving the generalization and robustness of recurrent neural networks. Amid the continual proposal of novel network models (Liu et al., 2024b; Ventola et al., 2023), enhancing network flexibility and adaptability through lightweight, model-agnostic hyper-networks has become increasingly valuable.

In probabilistic deep learning, Gaussian Process (GP) has emerged as a popular choice for both deep networks (Damianou & Lawrence, 2013) and lightweight surrogate modeling (Marrel & Iooss, 2024; Li et al., 2023). As a non-parametric Bayesian model, GPs offer strong prior integration and closed-form posterior updates, making them well-suited for meta-learning and hyper-network applications. For example, Patacchiola et al. (2020a); Sendera et al. (2021); Myers & Sardana (2021)

extend GP models by incorporating deep kernels, continuous normalizing flows, and variational latent variables. Wang et al. (2021) introduces class-specific latent variables tailored to meta-learning classification and Karaletsos & Bui (2020) adjusts GP for weight initialization for BNN.

# 3 PRELIMINARY

## 3.1 FEW-SHOT LEARNING

Few-shot learning refers to the ability to effectively learn and generalize from a limited number of training examples, often ranging from a single sample to a small handful of samples (Chen et al., 2019). In few-shot learning, the input-output pairs are grouped into a support set $\mathcal{D}_S = \{(x_i, y_i)\}_{i=1}^{K}$ and a query set $\mathcal{D}_Q = \{(x_i, y_i)\}_{i=1}^{M}$, with significant disparity in dataset size ($K \ll M$). The support and query sets together form a episode set $\mathcal{D} = (\mathcal{D}_S, \mathcal{D}_Q)$. From this perspective, the training dataset $\mathcal{D}_{\text{train}} = \{\mathcal{D}_t\}_{t=1}^{T}$ and test dataset $\mathcal{D}_{\text{test}} = (\mathcal{D}_S^*, \mathcal{D}_Q^*)$ can be viewed as independent and identically distributed (i.i.d.) subsets sampled from a shared distribution $p(\mathcal{D})$. A few-shot algorithm is trained on episodes $\mathcal{D}_t \sim \mathcal{D}_{\text{train}}$ and is expected to predict the membership of samples in the query set $\mathcal{D}_Q^*$ after model adaptation on the support set $\mathcal{D}_S^*$.

## 3.2 BAYESIAN META-LEARNING

**Variational Inference for Meta-Learning.** In the context of variational inference (VI), network parameters $\boldsymbol{\theta}$ are treated as stochastic variables following a prior distribution $p(\boldsymbol{\theta})$, and the posterior distribution $p(\boldsymbol{\theta} \mid \mathcal{D}) \propto p(\boldsymbol{\theta})p(\mathcal{D} \mid \boldsymbol{\theta})$ is optimized using Bayes' theorem. In practice, posterior distribution $p(\boldsymbol{\theta} \mid \mathcal{D})$ is approximated by a parameterized distribution $q(\boldsymbol{\theta})$ from a family of distributions. This forms the basis of Bayesian Neural Networks (BNNs), where the inference process involves integral over the parameter distribution, approximated using Monte Carlo sampling:

$$q(y \mid x) = \int p(y \mid x, \boldsymbol{\theta})q(\boldsymbol{\theta})d\boldsymbol{\theta} \approx \frac{1}{N}\sum_{i=1}^{N} p(y \mid x, \boldsymbol{\theta}_i), \tag{1}$$

where $\boldsymbol{\theta}_i \sim q(\boldsymbol{\theta})$.

To enable episodic-specific parametrization for meta-learning, hierarchical variational inference introduces global latent variables $\boldsymbol{\phi}$ and episode-specific parameter variables $\boldsymbol{\theta}_t$ for each episode $\mathcal{D}_t \sim \mathcal{D}_{\text{train}}$ (Amit & Meir, 2019). The output for a given episode is modeled as $p(y_i \mid x_i, \boldsymbol{\theta}_t)$, where $\boldsymbol{\theta}_t \sim p(\boldsymbol{\phi})$. The variational approximations for the global and episode-specific variables, $q(\boldsymbol{\phi}; \boldsymbol{\psi})$ and $q(\boldsymbol{\theta}_t; \boldsymbol{\lambda}_t)$, are parameterized by $\boldsymbol{\psi}$ and $\boldsymbol{\lambda}_t$, respectively. However, straightforward modeling of episode-specific variational parameters $\boldsymbol{\lambda}_t$ scales linearly with the number of training episodes, leading to high computational costs in environments with a large number of episodes and limited generalization to unseen episodes.

**Amortized Variational Meta-Learning.** A more efficient approach is to optimize the posterior distribution of parameters for each episode, $q_{\boldsymbol{\phi}}(\boldsymbol{\theta}_t \mid \mathcal{D}_t)$ (Ravi & Beatson, 2019). This method is built upon amortized variational inference, which establishes conditional dependence on input space to relief the learning burden in variational inference pipeline. The Amortized Bayesian Meta Learning (ABML)(Ravi & Beatson, 2019) extends previous study on prior adjustment by incorporating gradient descent calculation into episode-specific parameters generation. In this setup, only the shared initialization, parameterized by the variational approximation $q(\boldsymbol{\phi}; \boldsymbol{\psi})$, is updated to maximize the hierarchical Evidence Lower Bound (ELBO):

$$\mathcal{L}(\mathcal{D}_{1:T}) = \underset{q(\boldsymbol{\phi}; \boldsymbol{\psi})}{\boldsymbol{E}} \left[ \sum_{t=1}^{T} \underset{q(\boldsymbol{\theta}_t \mid \mathcal{D}_t)}{\boldsymbol{E}} [\log p(\mathcal{D}_t \mid \boldsymbol{\theta}_t)] - \text{KL}(q_{\boldsymbol{\phi}}(\boldsymbol{\theta}_t \mid \mathcal{D}_t) \| p(\boldsymbol{\theta}_t \mid \boldsymbol{\phi})) \right] - \text{KL}(q(\boldsymbol{\phi}; \boldsymbol{\psi}) \| p(\boldsymbol{\phi})). \tag{2}$$

## 3.3 PROBABILISTIC MODELING METHODS

**Gaussian Process(GP).** GP is a flexible non-parametric model widely used for probabilistic modeling (Seeger, 2004). In this framework, regression tasks are modeled as noisy observations on a limited dataset: $y_i = f(\boldsymbol{x}_i) + \epsilon_i$, where $\epsilon_i \sim \mathcal{N}(0, \sigma^2)$ and $\mathcal{D} = \{(\boldsymbol{x}_i, y_i)\}_{i=1}^{N}$. The latent function $f(\boldsymbol{x})$ follows a prior distribution $f(\boldsymbol{x}) \sim \mathcal{GP}(m(\boldsymbol{x}), k(\boldsymbol{x}, \boldsymbol{x}'))$ parameterized by mean function

$m(\boldsymbol{x})$ and kernel function $k(\boldsymbol{x}, \boldsymbol{x}')$. Considering the noise interference, outputs are modeled as $p(\boldsymbol{y} \mid \boldsymbol{f}) = \mathcal{N}(\boldsymbol{y} \mid \boldsymbol{f}, \sigma^2 I_N)$, where $\boldsymbol{f} = [f(\boldsymbol{x}_1), \ldots, f(\boldsymbol{x}_N)]^\top$.

Despite the flexibility, GPs are computationally expensive, with a time complexity of $\mathcal{O}(N^3)$. To address this, sparse variational Gaussian Process methods introduce inducing points $\boldsymbol{z} = \{z_1, \ldots, z_M\}$ ($M \ll N$) and corresponding latent function values $\boldsymbol{u} = \{u_1, \ldots, u_M\}$, connected with the modeled function $\boldsymbol{u} = f(\boldsymbol{z})$. These inducing variables summarize the dataset, reducing the computational complexity to $\mathcal{O}(NM^2)$. In addition to scalability improvement, inducing variables also enable parametrization of GPs, allowing them to adapt to training data without fully preserving fidelity.

**Normalizing Flow(NF).** NF is a class of generative model that construct complex, high-dimensional probability distributions by applying a sequence of invertible and differentiable transformations to a base distribution. With latent variable $\boldsymbol{z}_0 \sim \pi(\boldsymbol{z}_0)$, NF conducts the transformation on the base variable through a composition of K invertible functions $\boldsymbol{F}(\boldsymbol{z}_0) = \boldsymbol{F}_K \circ \cdots \circ \boldsymbol{F}_1(\boldsymbol{z}_0)$. This chain of transformation produces sampling log-likelihood specified as $\log q(\boldsymbol{F}(\boldsymbol{z}_0)) = \log \pi(\boldsymbol{z_0}) - \sum_{k=1}^{K} \log \left| \det \frac{\partial \boldsymbol{F}_k}{\partial \boldsymbol{z}_{k-1}} \right|$, where $\frac{\partial \boldsymbol{F}_k}{\partial \boldsymbol{z}_{k-1}}$ denotes the Jacobian matrix of the $k$-th transformation.

In the context of variational inference, NF can be used to construct highly flexible posterior approximations, moving beyond the limitations of simpler distributions like Gaussians and thereby improving the tightness of the evidence lower bound.

## 4 HIERARCHICAL PARAMETRIZATION WITH GAUSSIAN PROCESS FOR BAYESIAN META-LEARNING

### 4.1 GENERAL DERIVATION

The standard approach in amortized Bayesian meta-learning entails computing episode-specific posterior weights $q_{\boldsymbol{\phi}}(\boldsymbol{\theta}_t \mid \mathcal{D}_t)$ through gradient-based optimization, utilizing priors derived from the global latent variable $p(\boldsymbol{\theta}_t \mid \boldsymbol{\phi})$. However, in the context of few-shot learning, the limited size of the episode-specific support set often undermines the reliability of such optimization, leading to performance bottlenecks as the scale and number of episodes grow. To mitigate this issue, episode-specific posterior weights can be enhanced by integrating information from training samples beyond their respective episodes, thereby producing more informed weights conditioned on the input support set. This objective can be effectively realized through the introduction of a decoupling Gaussian Process for hierarchical parametrization, which enables the generation of episode-specific parameterizations via feed-forward computation. On this basis, a normalizing flow is also integrated, as a means to convert Gaussian distribution into a posterior more akin to the parameterization in actual neural network. Notably, this approach exhibits strong generalization interpretability and provides uncertainty estimates through approximate sampling, all while requiring minimal modifications to the existing network architecture.

We provide an illustration in Figure 1 to help understand this process. Given each input-output pair $(\boldsymbol{x}_{t,i}, \boldsymbol{y}_{t,i})$ from $t$-th task episode, the learning objective is to train a meta-Bayesian model comprising the induced variables $\boldsymbol{z}, \boldsymbol{u}$ for the Gaussian posterior $\boldsymbol{\omega}_t$ (Equation 3). Then, with the Gaussian posterior $\boldsymbol{\omega}_t$, we induce the IAF parameters $\boldsymbol{w}$ for the deep network's posterior $\boldsymbol{\theta}_t$ (Equation 4), which is leveraged for final prediction. This thereby enables the model to approximate the complex posterior distribution for meta-learning. For meta-training scheme, the training process is repeated for each task episode $\mathcal{D}_t$. For meta-testing scheme, given support set and query set, HP-GP generates the Gaussian posterior $\hat{\boldsymbol{\omega}}$, and then the deep network's posterior $\hat{\boldsymbol{\theta}}$ for prediction. The deduction details are as follows:

Under the framework of HP-GP, the weight distribution is defined with a Sparse Variational Gaussian Process (SVGP) prior: $p(\boldsymbol{\omega}, \boldsymbol{u}; \boldsymbol{x}, \boldsymbol{z}) = \mathcal{N}(\boldsymbol{\omega} \mid \boldsymbol{K}_{XZ} \boldsymbol{K}_{ZZ}^{-1} \boldsymbol{u}, \boldsymbol{K}_{XX} - \boldsymbol{K}_{XZ} \boldsymbol{K}_{ZZ}^{-1} \boldsymbol{K}_{ZX}) \mathcal{N}(\boldsymbol{u} \mid \boldsymbol{0}, \boldsymbol{K}_{ZZ})$, where $\boldsymbol{u}$ and $\boldsymbol{\omega}$ are inducing function values and design function values, and $\boldsymbol{z}$ and $\boldsymbol{x}$ are the corresponding inducing locations and design locations (Salimbeni & Deisenroth, 2017). Unlike traditional settings where the global parameters are expected to provide priors for episode-specific parameters $\boldsymbol{\omega}_t$, they are instead exploited to facilitate posterior assimilation over the specific input

$\boldsymbol{x}_t$:

$$q(\boldsymbol{\omega}_{x_t} \mid \boldsymbol{x}_t, \boldsymbol{z}) = \int p(\boldsymbol{\omega}_{x_t} \mid \boldsymbol{u}; \boldsymbol{x}_t, \boldsymbol{z}) q(\boldsymbol{u}) d\boldsymbol{u}, \tag{3}$$

where $q(\boldsymbol{u}) = \mathcal{N}(\boldsymbol{u} \mid \boldsymbol{m}, \boldsymbol{S})$ is the posterior of variational latent function and $q(\boldsymbol{\omega}_{x_t} \mid \boldsymbol{x}_t, \boldsymbol{z}) = \mathcal{N}(\boldsymbol{\omega}_{x_t} \mid \boldsymbol{\mu}(\boldsymbol{x}_t), \boldsymbol{\Sigma}(\boldsymbol{x}_t))$. To ensure lightweight and model-agnostic implementation, we propose a decoupling realization, which will be detailed in Section 4.2.

Afterwards, the weight posterior is output via NF pipeline $\boldsymbol{\theta}_t = \boldsymbol{F}(\boldsymbol{\omega}_t)$, fitting the actual weight posterior based on Gaussian distribution suggestion:

$$\boldsymbol{\mu}_t = \underset{x_t \sim \mathcal{D}_t}{E} \left[ \boldsymbol{\mu}(x_t) \right], \boldsymbol{\Sigma}_t = \underset{x_t \sim \mathcal{D}_t}{E} \left[ \boldsymbol{\Sigma}(x_t) + (\boldsymbol{\mu}(x_t) - \boldsymbol{\mu}_t)^\top (\boldsymbol{\mu}(x_t) - \boldsymbol{\mu}_t) \right],$$

$$q(\boldsymbol{\theta}_t \mid \mathcal{D}_t) = q(\boldsymbol{F}^{-1}(\boldsymbol{\theta}_t) \mid \mathcal{D}_t, \boldsymbol{z}) \left| \frac{\partial \boldsymbol{F}^{-1}(\boldsymbol{\theta}_t)}{\partial \boldsymbol{\theta}_t} \right|, q(\boldsymbol{\omega}_t \mid \mathcal{D}_t, \boldsymbol{z}) = \mathcal{N}(\boldsymbol{\omega}_t \mid \boldsymbol{\mu}_t, \boldsymbol{\Sigma}_t). \tag{4}$$

The variational posterior for episode-specific latent function is calculated via maximum entropy principle over $\boldsymbol{x}_t \sim \mathcal{D}_t$ (Proposition 1), providing episode-specific distribution summation with minimal external hypothesis.

The combined integration of SVGP and NF into hierarchical variational inference pipeline provides two key benefits for meta-learning: 1) SVGP enables incremental assimilation of episode-specific parameters, avoiding the challenges of parameter scaling or catastrophic forgetting. 2) NF outputs posterior parametrization analogous to network parameters distributions, allowing seamlessly represent episode-specific parameters with knowledge gained from both support set and query set.

With the integration of SVGP, the objective to maximize $\log(p(\mathcal{D}))$ leads to the following ELBO:

$$\log[\prod_{t=1}^{T} p(\mathcal{D}_t)] = \log[\prod_{t=1}^{T} \int p(\mathcal{D}_t \mid \boldsymbol{\theta}_t) p(\boldsymbol{\theta}_t) d\boldsymbol{\theta}_t] \geq \sum_{t=1}^{T} \underset{q(\boldsymbol{\theta_t} \mid \mathcal{D}_t)}{\boldsymbol{E}} [\log p(\mathcal{D}_t \mid \boldsymbol{\theta}_t)] - \mathrm{KL}(q(\boldsymbol{\theta}_t \mid \mathcal{D}_t) \| p(\boldsymbol{\theta}_t)). \tag{5}$$

For computational convenience, KL divergence term $\sum_{t=1}^{T} \mathrm{KL}(q(\boldsymbol{\theta}_t \mid \mathcal{D}_t) \| p(\boldsymbol{\theta}_t))$ can be further simplified as illustrated in Proposition 2 in Appendix. The nature of NF ensures that $\mathrm{KL}(q(\boldsymbol{\theta}_t \mid \mathcal{D}_t) \| p(\boldsymbol{\theta}_t)) = \mathrm{KL}(q(\boldsymbol{\omega}_t \mid \mathcal{D}_t, \boldsymbol{z}) \| p(\boldsymbol{\omega}_t \mid \boldsymbol{u}; \mathcal{D}_t, \boldsymbol{z}))$. Considering that the prior and variational posterior of global variational variables $\boldsymbol{u}$ and $\boldsymbol{z}$ determines corresponding parameterization distribution $\boldsymbol{\theta}_t$ via conditional dependence, the regularization term on the latent variable prediction can be further replaced by a regularization term on the global variational variables, which reduces repetitive calculations across episodes. The final ELBO thus becomes:

$$\mathcal{L}(\mathcal{D}_{1:T}) = \sum_{t=1}^{T} \underset{q(\boldsymbol{\theta_t} \mid \mathcal{D}_t)}{\boldsymbol{E}} [\log p(\mathcal{D}_t \mid \boldsymbol{\theta}_t)] - \mathrm{KL}(q(\boldsymbol{u}) \| p(\boldsymbol{u}; \boldsymbol{z})). \tag{6}$$

## 4.2 DECOUPLING REALIZATION

A standard approach to implementing the aforementioned SVGP for episode-specific probabilistic parametrization involves designing a episode-specific hyper-network to assimilate inputs directly, as demonstrated in Gordon et al. (2019). However, to ensure lightweight and model-agnostic implementation, we adopt a layer-wise implementation of the amortized SVGP module. To further reduce the scale of latent parameters and improve computation efficiency during SVGP calculation, we adopt a decoupling realization of GP calculation, inspired by preceding attempts to streamline intermediate variables of the model (Hu et al., 2022). The weight tensor is subdivided into input and output component, connected by tensor product to produce formal GP prediction. An Inverse Autoregressive Flow (Kingma et al., 2016) is modified for distribution conversion to avoid performance worsening induced by vanilla Gaussian posterior in standard BNNs (Wenzel et al., 2020; Izmailov et al., 2021). Within the IAF implementation, a Kolmogorov-Arnold Network (Liu et al., 2024a) transformation is applied to increase parameterization efficiency and the fitting effect of distribution transformation.

We start with a $L$-layer Neural Network provided as a cascade of transformations:

$$\boldsymbol{x}^{(l)} = f^{(l)}(\boldsymbol{x}^{(l-1)}; \boldsymbol{\theta}^{(l)}), l = 1, \dots, L, \boldsymbol{x}^{(0)} = \boldsymbol{x}, x^{(L)} = y, \tag{7}$$

where $\boldsymbol{x}^{(l)}$ denotes the output of the $l$-th layer, and $\boldsymbol{\theta}^{(l)}$ represents the parameters of the $l$-th layer.

The network parameter tensor can be represented by the product of input-output tensor components, specifically defined by considering the parameterization origin of each component of the tensor, $\boldsymbol{\theta}^{(l)} = \boldsymbol{\theta}^{(l)}_{(\text{in})} \otimes \boldsymbol{\theta}^{(l)}_{(\text{out})}$. This factorization provides the coupling calculation pipeline for the SVGP model, which adopts a variable factorization with identical structure $\boldsymbol{\omega}^{(l)} = \boldsymbol{\omega}^{(l)}_{(\text{in})} \otimes \boldsymbol{\omega}^{(l)}_{(\text{out})}$.

In the case of practical calculation, the computation of the two components precedes the tensor product as follows:

$$
\begin{aligned}
\boldsymbol{\mu}^{(l)}_{(d)}(\boldsymbol{x}^{(l-1)}_t) &= (\boldsymbol{\alpha}^{(l)}_{(d)})^{\top} \boldsymbol{m}^{(l)}_{(d)}, \\
\boldsymbol{\Sigma}^{(l)}_{(d)}(\boldsymbol{x}^{(l-1)}_t) &= k^{(d,l)}_{xx} - (\boldsymbol{\alpha}^{(l)}_{(d)})^{\top}(\boldsymbol{K}^{(d,l)}_{zz} - \boldsymbol{S}^{(l)}_{(d)})\boldsymbol{\alpha}^{(l)}_{(d)}, \\
\boldsymbol{\alpha}^{(l)}_{(d)} &= (\boldsymbol{K}^{(d,l)}_{zz})^{-1} k^{(d,l)}_{zx},
\end{aligned}
\tag{8}
$$

where $d \in \{in, out\}$ and $k^{(d,l)}_{zx} = [k^{(d,l)}(\boldsymbol{x}^{(l-1)}_t, z^{(l-1)}_1), \ldots, k^{(d,l)}(\boldsymbol{x}^{(l-1)}_t, z^{(l-1)}_M)]^{\top}$.

The tensor product on mean and kernel variables produces the eventual posterior distribution $q(\boldsymbol{\omega}^{(l)}_{x_t} \mid \boldsymbol{x}^{(l-1)}_t, \boldsymbol{z}^{(l-1)})$, with specific details deducted in Proposition 3:

$$
\begin{aligned}
q(\boldsymbol{\omega}^{(l)}_{x_t} \mid \boldsymbol{x}^{(l-1)}_t, \boldsymbol{z}^{(l-1)}) &= \mathcal{N}(\boldsymbol{\omega}^{(l)}_{x_t} \mid \boldsymbol{\mu}^{(l)}(\boldsymbol{x}^{(l-1)}_t), \boldsymbol{\Sigma}^{(l)}(\boldsymbol{x}^{(l-1)}_t)), \\
\boldsymbol{\mu}^{(l)}(\boldsymbol{x}^{(l-1)}_t) &= \boldsymbol{\mu}^{(l)}_{\text{in}}(\boldsymbol{x}^{(l-1)}_t) \otimes \boldsymbol{\mu}^{(l)}_{\text{out}}(\boldsymbol{x}^{(l-1)}_t), \\
\boldsymbol{\Sigma}^{(l)}(\boldsymbol{x}^{(l-1)}_t) &= \boldsymbol{\Sigma}^{(l)}_{\text{in}}(\boldsymbol{x}^{(l-1)}_t) \otimes [\boldsymbol{\mu}^{(l)}_{\text{out}}(\boldsymbol{x}^{(l-1)}_t)]^2 + [\boldsymbol{\mu}^{(l)}_{\text{in}}(\boldsymbol{x}^{(l-1)}_t)]^2 \otimes \boldsymbol{\Sigma}^{(l)}_{\text{out}}(\boldsymbol{x}^{(l-1)}_t) + \boldsymbol{\Sigma}^{(l)}_{\text{in}}(\boldsymbol{x}^{(l-1)}_t) \otimes \boldsymbol{\Sigma}^{(l)}_{\text{out}}(\boldsymbol{x}^{(l-1)}_t),
\end{aligned}
\tag{9}
$$

where $\otimes$ is the tensor product and abbreviation $[\boldsymbol{\mu}^{(l)}_{(d)}(\boldsymbol{x}^{(l-1)}_t)]^2 = \boldsymbol{\mu}^{(l)}_{(d)}(\boldsymbol{x}^{(l-1)}_t) \otimes \boldsymbol{\mu}^{(l)}_{(d)}(\boldsymbol{x}^{(l-1)}_t)$.

With element-specific intermediate variable $\boldsymbol{\mu}(\boldsymbol{x}_t)$ and $\boldsymbol{\Sigma}(\boldsymbol{x}_t)$ available, an accumulation is necessary for its conversion into episode-specific variables $q(\boldsymbol{\omega}_t \mid \mathcal{D}_t, \boldsymbol{z})$, as described in Equation 4. The outcome is processed with the Inverse Autoregressive Flow(IAF), a normalizing flow built on the invertible and differentiable transformation MADE (Germain et al., 2015), featuring parallel computation on latent input and adaptability to high-dimensional distribution fitting with limited assumptions on the arrangement of feature components. We consider KAN as the transformation realization for $f_{\beta^i}$ and $f_{\alpha^i}$, in an attempt to excel the performance of a traditional MLP component with minimized layers and computation burden. An ablation study on the utility of this design with comparison to MLP implementations is also provided in Table 4 in Appendix for deeper understanding of the network architectural design. The resulting calculation can be formulated as:

$$
\theta^{i,(l)}_t = \omega^i_{t,(l)} \exp \alpha^i + \beta^i, \quad \beta^i = f_{\beta^i}(\boldsymbol{\omega}^{1:i-1}_{t,(l)}), \quad \alpha^i = f_{\alpha^i}(\boldsymbol{\omega}^{1:i-1}_{t,(l)}), \quad f_j(\boldsymbol{\omega}^{1:i-1}_{t,(l)}) = \boldsymbol{w}^{(l)}_i(\text{Act}(\boldsymbol{\omega}^{1:i-1}_{t,(l)}) + \text{Spline}(\boldsymbol{\omega}^{1:i-1}_{t,(l)})),
\tag{10}
$$

where $\boldsymbol{\theta}^{(l)}_t = [\theta^{1,(l)}_t, \ldots, \theta^{i,(l)}_t, \ldots, \theta^{D,(l)}_t]$, $j \in \{\alpha^i, \beta^i\}$, $\boldsymbol{w}_i$ is the output weight of KAN, and Act and Spline are commonly implemented with SiLU activation and B-spline transformation respectively (Singh et al., 2023; Gordon & Riesenfeld, 1974).

The above deduction provides detailed description for weight generation procedure through a simple feed-forward calculation. The decoupling design reduces the time complexity for SVGP computation down to $\mathcal{O}(((d^{(l)}_{(in)})^3 + (d^{(l)}_{(out)})^3)m^3)$ ($m = |\boldsymbol{z}|$ denotes number of inducing points), significantly alleviating the computational burden exacerbated by the parameter scaling in modern deep neural networks. For both meta-training and meta-testing, $\boldsymbol{\theta}_t$ is adopted by the base model for downstream inference. In the meta-testing phase, the key distinction is that both support and test set samples can be utilized for weight generation, fully leveraging the abundance of available adaptation samples.

### 4.3 Theoretical Analysis

In this section, we provide our theoretical analysis for HP-GP on a theoretical modeling on standard feed-forward neural network component. We bridge the Reproducing Kernel Hilbert Space (RKHS) for analyzing Gaussian processes with the Barron space framework, a theoretical framework tailored for ideal Bayesian networks analysis, to establish a unified theoretical foundation. The analysis is

commenced with the norm bound calculation of SVGP output, before moving forward to assertions of generalization capacity measured by Rademacher Complexity and approximation bias of Monte Carlo sampling of the network, crucial for guarantee of performance stability for actual rollout.

The following theorems rely on the definition of RKHS Space and Barron Space, which are introduced in Appendix A.2.

**Theorem 1.** *Assuming that input domain $\mathcal{X}$ is compact and $k$ is continuous, the regularization of inducing variables is bounded by $KL[q(\boldsymbol{u}^{(d)})\|p(\boldsymbol{u}^{(d)};\boldsymbol{z}^{(d)})] \leq B_{KL}, d \in \{in, out\}$, and the power of episode $|\mathcal{D}_t| < \infty$. With probability $\geq (1-\delta)^2$, the following assertion holds for HP-GP output based on bounded inputs $\|x_t\|_1 \leq 1$:*

$$\|\boldsymbol{\theta}_t\|_1 \leq 2D \max_j \sum_{j=1}^{D} |w_j|(1+L_j)[\frac{d}{\sqrt{\delta}}(8C_2^2|\mathcal{D}_t|(\max_i \|\phi_i\|_{L_\infty(\mathcal{D}_t)})^2 B_{KL}$$

$$+ C_1 \max_i \|\phi_i\|_{L_\infty(\mathcal{D}_t)}^4 + C_0 B_{KL}) + 2C_2^2|\mathcal{D}_t| \max_i \|\phi_i\|_{L_\infty(\mathcal{D}_t)}^2 B_{KL}]^2,$$

*where $C_0, C_1, C_2$ are constants, $\dim(\boldsymbol{\omega}_t) = D$, $k(\boldsymbol{x}, \boldsymbol{x}) = \sum_{i=1}^{\infty} \lambda_i \phi_i(\boldsymbol{x})^2$ with assumption $\sum_{i=1}^{\infty} \lambda_i^p < \infty, \forall p \in (0,1)$ $\boldsymbol{w}$ is the weight parameter, and spline function is assumed with Lipchitz continuity with Lipchitz constant $|Spline_j(x)| \leq L_j|x|$.*

**Theorem 2.** *(Rademacher complexity bound) With all assumptions in Theorem 1 hold, for the function class $\mathcal{F}_Q = \{f_{\boldsymbol{\omega}_t} \in \mathcal{B}\}$, and external output weight$\omega_{t,0}^{(2)}$, with probability $\geq (1-\delta)^2$, the empirical Rademacher complexity satisfies:*

$$\widehat{\mathfrak{R}}_n(\mathcal{F}_Q) \leq 4|\omega_{t,0}^{(2)}|D \max_j \sum_{j=1}^{D} |w_j|(1+L_j)[\frac{d}{\sqrt{\delta}}(8C_2^2|\mathcal{D}_t|(\max_i \|\phi_i\|_{L_\infty(\mathcal{D}_t)})^2 B_{KL}$$

$$+ C_1 \max_i \|\phi_i\|_{L_\infty(\mathcal{D}_t)}^4 + C_0 B_{KL}) + 2C_2^2|\mathcal{D}_t| \max_i \|\phi_i\|_{L_\infty(\mathcal{D}_t)}^2 B_{KL}]^2 \sqrt{\frac{2\ln(2d)}{|\mathcal{D}_t|}}.$$

**Theorem 3.** *(Approximation Error) With all assumptions in Theorem 1 hold, for the function class $\mathcal{F}_Q = \{f_{\boldsymbol{\omega}_t} \in \mathcal{B}\}$, with probability $\geq (1-\delta)^2$, there exists a standard neural network$f_{m,\boldsymbol{\Omega}}(\boldsymbol{x}) = \frac{1}{m}\sum_{k=1}^{m} \boldsymbol{\Omega}_{k,0}^{(2)} \sigma((\boldsymbol{\Omega}_{k,0}^{(1)})^\top \boldsymbol{x} + \boldsymbol{\Omega}_{k,1}^{(2)})$ that satisfies*

$$\|f(\cdot) - f_{m,\boldsymbol{\Omega}}(\cdot)\|^2 \leq \frac{12}{m}|\omega_{t,0}^{(2)}|^2 D^2 [\max_j \sum_{j=1}^{D} |w_j|(1+L_j)]^2 [\frac{d}{\sqrt{\delta}}(8C_2^2|\mathcal{D}_t|(\max_i \|\phi_i\|_{L_\infty(\mathcal{D}_t)})^2 B_{KL}$$

$$+ C_1 \max_i \|\phi_i\|_{L_\infty(\mathcal{D}_t)}^4 + C_0 B_{KL}) + 2C_2^2|\mathcal{D}_t| \max_i \|\phi_i\|_{L_\infty(\mathcal{D}_t)}^2 B_{KL}]^4.$$

The complete proof of the assertions above can be found in Appendix A.2 in supplementary materials.

## 5 EXPERIMENTS

In this section, we compare our method with several competitive baseline methods, including Deep Kernel Transfer(DKT, Patacchiola et al. (2020b)), Non-Gaussian Gaussian Processes (NGGP, Sendera et al. (2021)), Variational Meta-Gaussian Process(VMGP, Myers & Sardana (2021)), Scalable Meta-Learning with Gaussian Processes(ScaML-GP, Tighineanu et al. (2024)) and optimization-based meta-learning approaches, such as Model-Agnostic Meta-Learning(MAML, Finn et al. (2017)), Amortized Bayesian Meta-Learning(ABML, Ravi & Beatson (2019)), and Bayesian Model-Agnostic Meta-Learning(BMAML, Kim et al. (2018)). The comparison is conducted on two challenging meta-learning tasks:time series prediction and object pose prediction. In addition, an extensive ablation studies and visualization analyzes are also incorporated in the Appendix A.4 to further demonstrate the effectivess of the proposed method. Next, we will introduce our experiment results.

### 5.1 TIME SERIES PREDICTION

To evaluate the performance of HP-GP on real-world time series datasets, we conduct experiments on two widely recognized datasets: the NASDAQ dataset (Qin et al., 2017) and the EEG dataset

(Fraga & de Querétaro México. n esq. Mariano Escobedo, 2018). The NASDAQ dataset comprises 81 major stocks from the NASDAQ 100 index, while the EEG dataset records brainwave signals sampled at 128Hz across a large cohort of patients. To comprehensively assess the assimilation of support set information for domain generalization, we establish both *in-range* and *out-of-range* scenarios for the pivotal series data and their corresponding counterparts.

For the *in-range* prediction, we focus on the NASDAQ NDX100 index and the EEG patient's data from A001SB1_1. In this case, the first 70% of data points are used for meta-training, and the remaining 30% are used for meta-testing. In contrast, the *out-of-range* scenario involves extracting the target time series from the entire dataset.

We apply HP-GP to a three-layer MLP architecture and compare the performance in Table 1. The experimental results demonstrate the superior performance of HP-GP with linear kernel across both NASDAQ and EEG datasets. HP-GP achieves the lowest RMSE in all evaluation scenarios, notably reducing in-range prediction error on NASDAQ to $0.063\pm0.026$. Its out-of-distribution generalization capability is particularly remarkable, achieving $0.084$ RMSE $\pm$ $0.059$ on EEG out-of-range predictions – 3.9× lower than ABML's $0.366$ RMSE. The method exhibits exceptional stability, as evidenced by consistently small standard deviations ($\leq 0.06$ across all tests), outperforming DKT's maximum variability of $0.155$. These results collectively establish HP-GP as a robust solution for few-shot regression with enhanced accuracy and domain adaptability.

Table 1: Average Root Mean-Squared Error (RMSE) with standard deviation for few-shot regression on NASDAQ and EEG datasets (Over 10 trials, lowest error in bold.)

| METHODS | NASDAQ | | EEG | |
|---|---|---|---|---|
| | IN-RANGE($\downarrow$) | OUT-OF-RANGE($\downarrow$) | IN-RANGE($\downarrow$) | OUT-OF-RANGE($\downarrow$) |
| MAML(FINN ET AL., 2017) | $0.514_{\pm0.174}$ | $0.531_{\pm0.158}$ | $0.604_{\pm0.143}$ | $0.617_{\pm0.174}$ |
| ABML(RAVI & BEATSON, 2019) | $0.394_{\pm0.169}$ | $0.415_{\pm0.088}$ | $0.422_{\pm0.217}$ | $0.366_{\pm0.208}$ |
| ALPACA(HARRISON ET AL., 2018) | $0.323_{\pm1.647}$ | $0.332_{\pm1.359}$ | $0.464_{\pm1.464}$ | $0.422_{\pm1.648}$ |
| SCAML-GP(TIGHINEANU ET AL., 2024) | $0.454_{\pm0.055}$ | $0.343_{\pm0.183}$ | $0.188_{\pm0.249}$ | $0.138_{\pm0.051}$ |
| VMGP(MYERS & SARDANA, 2021) | $1.802_{\pm1.934}$ | $2.411_{\pm1.971}$ | $0.513_{\pm0.613}$ | $1.344_{\pm0.156}$ |
| NGGP(SENDERA ET AL., 2021) | $0.117_{\pm0.142}$ | $0.133_{\pm0.130}$ | $0.124_{\pm0.173}$ | $0.183_{\pm0.144}$ |
| DKT(PATACCHIOLA ET AL., 2020B) | $0.166_{\pm0.155}$ | $0.186_{\pm0.139}$ | $0.180_{\pm0.131}$ | $0.179_{\pm0.119}$ |
| **HP-GP(RQ)** | $0.066_{\pm0.031}$ | $\mathbf{0.069}_{\pm0.035}$ | $0.086_{\pm0.057}$ | $0.088_{\pm0.054}$ |
| **HP-GP(LINEAR)** | $\mathbf{0.063}_{\pm0.026}$ | $0.075_{\pm0.044}$ | $\mathbf{0.077}_{\pm0.049}$ | $\mathbf{0.084}_{\pm0.059}$ |
| **HP-GP(MATERN-0.5)** | $0.072_{\pm0.029}$ | $0.076_{\pm0.040}$ | $0.092_{\pm0.053}$ | $0.095_{\pm0.057}$ |

Additionally, a visualization of the self-correlation structure of Gaussian distribution output and posterior weight distribution is shown in Figure 3. We also explore the effect of hyper-network parametrization by altering kernel functions and number of inducing variables of the hyper-network, and the experimental outcome is reported in Appendix A.3.1.

## 5.2 OBJECT POSE PREDICTION

The object pose prediction dataset (Yin et al., 2020) provides a formal benchmark for the memorization problem evaluation on meta-learning models, by constructing a non-mutually-exclusive

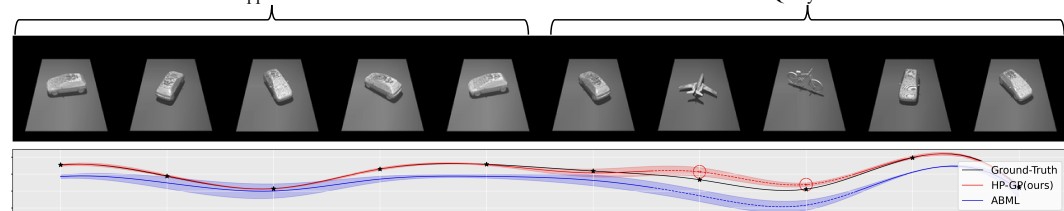

Figure 2: In non-mutually exclusive regression tasks, HP-GP stores cross-class information within its inducing variables, providing episode-specific memorization instead of unified memorization that is fatal to meta-testing generalization. Under the validation scenario where support set consists of automobile images, HP-GP delivers reasonable predictions for airplane and bicycle images in the query set (red circle). Furthermore, the uncertainty estimation for erroneous inputs reveals the model's perplexity, offering precise uncertainty calibration for out-of-domain applications.

regression task. The dataset is consisting of grayscale images of object models from 10 classes, each with a 10 images with resolution of $128 \times 128$ and random orientations for regression. This challenging mission is designed to corrupt test models' generalization capabilities by encouraging memorization of canonical orientations in a meta-training set consisting of limited training episodes. A over-parameterized meta-learning model is prone to establish a unified prediction pipeline on meta-testing samples and fails to effectively exploiting the appearance information of support set samples, rendering significantly worsened meta-testing performance.

Despite the inefficacy of traditional over-fitting counter-measures such as regularization and dropout techniques, the challenge of non-mutually exclusive datasets can be smoothly resolved by the non-parametric nature of the probabilistic modeling methods applied, as illustrated in Figure 2. Notably, in this visualization, the convolutional weights are calculated on individual inputs instead of episodes, which is unconventional but necessary to demonstrate the resilience of HP-GP under this scenario. The GP amortization network effectively captures cross-class information via inducing variables, enabling the model to grasp useful samples from both support and query sets while avoiding the impact of potential mismatch.

Two base models are selected for this experiment: a small 4-layer CNN with 3-layer MLP regressor and a standard Resnet-18. The formal backbone is chosen primarily in consideration of maintaining comparison fairness with predeceasing SoTA models on the dataset (Sendera et al., 2021), which are also built on this network as their backbone. The HP-GP is applied to the first convolutional layers in both 4-layer Convolution backbone and a standard ResNet-18 backbone. We compare the HP-GP method with other prominent meta-learning approaches, and report the results in Table 2.

According to the experiment outcome, HP-GP demonstrates state-of-the-art performance on few-shot pose prediction task, achieving simultaneous improvements across all three evaluation metrics. When combined with RBF kernel on Conv4 backbone, HP-GP attains the lowest MSE of $0.943\pm0.219$ - a 57% reduction compared to the previous best method DKT ($2.148 \pm 0.426$). Notably, HP-GP achieves this while maintaining superior correlation (PCC=$0.943 \pm 0.029$) and coverage rate (CR=$0.711 \pm 0.279$), indicating both accurate predictions and reliable uncertainty quantification. This breakthrough performance suggests HP-GP's hierarchical parameterization architecture effectively captures geometric relationships in few-shot regimes, overcoming limitations of conventional meta-learning approaches that suffer from error propagation (e.g., BMAML's 8.015 MSE) or weak correlation modeling (ABML's 0.456 PCC). In contrast, ResNet-18 exhibits a significantly worse performance on the task, suggesting the challenge of memorization problem on non-exclusive regression task exacerbated by over-parameterization in modern DNN structures (Yin et al., 2020). Even on worsened baseline, ResNet-18 enhanced by HP-GP provides a significantly lowered MSE from $9.092\pm1.463$ to $6.904\pm2.227$, and non-trivial CR metric valuable for downstream confidence estimation on model predictions.

Table 2: Average Mean-Squared Error (MSE), Pearson Correlation Coefficient (PCC) and Coverage Rate (CR) with standard deviation for few-shot object pose regression (Over 5 trials, best performance in bold, * denotes values reported in Yin et al. (2020).)

| METHODS | MSE($\downarrow$) | PCC($\uparrow$) | CR($\uparrow$) |
|---|---|---|---|
| FEATURE TRANSFER* | $7.330_{\pm0.350}$ | - | - |
| MAML*(FINN ET AL., 2017) | $5.390_{\pm1.310}$ | - | - |
| MR-MAML*(YIN ET AL., 2020) | $2.260_{\pm0.090}$ | - | - |
| CNP*(GARNELO ET AL., 2018) | $8.480_{\pm0.120}$ | - | - |
| BMAML(KIM ET AL., 2018) | $8.015_{\pm2.241}$ | $0.525_{\pm0.389}$ | $0.040_{\pm0.089}$ |
| ABML(RAVI & BEATSON, 2019) | $9.402_{\pm3.484}$ | $0.456_{\pm0.129}$ | $0.080_{\pm0.110}$ |
| DKT(PATACCHIOLA ET AL., 2020B) | $2.148_{\pm0.426}$ | $0.714_{\pm0.385}$ | $0.453_{\pm0.142}$ |
| NGGP(SENDERA ET AL., 2021) | $2.326_{\pm0.215}$ | $0.665_{\pm0.286}$ | $0.363_{\pm0.213}$ |
| **VANILLA RESNET18** | $9.092_{\pm1.463}$ | $0.116_{\pm0.314}$ | $0_{\pm0}$ |
| **HP-GP + RESNET18(RBF)** | $7.079_{\pm2.307}$ | $0.749_{\pm0.270}$ | $0.562_{\pm0.278}$ |
| **HP-GP + RESNET18(LINEAR)** | $6.904_{\pm2.227}$ | $0.736_{\pm0.301}$ | $0.420_{\pm0.229}$ |
| **VANILLA CONV4** | $5.758_{\pm2.524}$ | $0.571_{\pm0.439}$ | $0_{\pm0}$ |
| **HP-GP + CONV4(RBF)** | $\mathbf{0.982}_{\pm0.219}$ | $\mathbf{0.943}_{\pm0.029}$ | $\mathbf{0.711}_{\pm0.279}$ |
| **HP-GP + CONV4(LINEAR)** | $1.172_{\pm0.654}$ | $0.928_{\pm0.044}$ | $0.681_{\pm0.353}$ |

## 6 CONCLUSIONS

In this work, we propose a GP-based Bayesian learning framework to enhance the performance of deep networks in the context of meta-learning. Our method is implemented layer-wise and episodic scalable, offering a lightweight and model-agnostic alternative for realizing Bayesian meta-learning. The hierarchical decoupling GP structure with IAF boosting effectively captures shifts in network weight distribution in response to changes in input distribution, thus preparing the latent variable space for episodic-specific parameter generation.

The proposed framework demonstrates competitive efficacy compared to established meta-learning baselines, as evidenced by atypical data inconsistency challenges including domain shift and memorization problems. However, there exists certain limitations for our method, which does not extend to the modern large language models (LLMs) due to limited computational resources. We leave it for future work to explore extending HP-GP for meta learning in LLM to enhance its reasoning ability (Liu et al., 2025).

Therefore, to make every effort to overcome the limitations of HP-GP and further expand its scope of application, our following works will keep on enhancing hyper-parameterization automation for modern large-scale deep learning framework through the two relevant pathways: (1) Discover the systematic identification of optimal intervention layers of heterogeneous deep network via gradient-based feature impact analysis, (2) development of hybrid kernel mechanisms that generates appropriate scalable kernel to minimize computational complexity while preserving representational capacity and validity amid episode shift, and (3) extension of the proposed algorithmic paradigm to modern LLMs.

**Ethics Statement**

This work complies with the ICLR Code of Ethics. No ethical concerns requiring specific disclosure are present.

Large Language Models are used to aid and polish writing in this work. The usage is limited to sentence polishing, error checking, Latex formatting, and improvement of fluency and clarity of the manuscript.

**Reproducibility Statement**

We have made every effort to ensure the reproducibility of our work and have provided detailed disclosure materials, including the following:

(1) Detailed introduction on the kernel functions referred in Appendix A.1.

(2) Detailed deduction of propositions and full proof of theorems in Appendix A.2. The analysis is built upon the theoretical achievements of preceding works (Kanagawa et al., 2018; Zhang, 2023), to maintain rigorousness and accuracy of analysis.

(3) Implementation details of experiments in Appendix A.3. The evaluation protocols follow the standard specification of preceding works (Patacchiola et al., 2020b; Sendera et al., 2021), to maintain comparison fairness and reproducibility convenience.

(4) Extra experiments and ablation analysis of the proposed method in Appendix A.4.

(5) Source code for training and inference during the work upon paper acceptance.

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

# A APPENDIX

## A.1 KERNEL FUNCTIONS

In this section, we make detailed descriptions on the kernel functions used in our experiments for the convenience of replication, which are central to modeling the covariance structure of the data in a Gaussian Process framework.

### A.1.1 RADIAL BASIS FUNCTION (RBF) KERNEL

The Radial Basis Function (RBF) kernel, also known as the Gaussian kernel, is a stationary kernel that measures the similarity between two input points based on the squared Euclidean distance between them. The kernel is defined as:

$$k(x, x') = \exp\left(-\frac{\|x - x'\|^2}{2l^2}\right)$$

where $l$ is a lengthscale parameter, learned during training. This kernel is widely utilized in Gaussian Processes for its ability to model smooth, non-linear functions, making it suitable for a variety of regression and classification tasks.

### A.1.2 COSINE SIMILARITY KERNEL

The Cosine Similarity kernel measures the cosine of the angle between two input vectors, providing a similarity measure that ranges from -1 to 1. It is computed as the normalized inner product of the input vectors:

$$k(x, x') = \frac{x^T x'}{\|x\|\|x'\|}$$

where $\|x\|$ and $\|x'\|$ denote the Euclidean norms of the input vectors $x$ and $x'$, respectively. This kernel is particularly useful for tasks where the relative orientation of the input vectors is of primary interest, rather than their absolute magnitude.

### A.1.3 LINEAR KERNEL

The Linear kernel is a fundamental kernel function that computes the similarity between two input vectors via their dot product. It is defined as:

$$k(x, x') = x^T x' + c$$

where $c$ is an optional constant term that introduces a bias adjustment, enhancing the model's flexibility by allowing an offset from the origin. This kernel inherently assumes a linear relationship between the input variables and is computationally efficient due to its simplicity.

### A.1.4 RATIONAL QUADRATIC (RQ) KERNEL

The Rational Quadratic (RQ) kernel is a flexible kernel that can be interpreted as an infinite sum of Radial Basis Function (RBF) kernels with varying length scales. Its mathematical formulation is:

$$k(x, x') = \left(1 + \frac{\|x - x'\|^2}{2\alpha l^2}\right)^{-\alpha}$$

where $\alpha > 0$ is the scale-mixture parameter (also known as the shape parameter), and $l > 0$ is the characteristic length scale. The parameter $\alpha$ controls the weighting of different scales of variation: smaller values of $\alpha$ result in heavier tails, accommodating longer-range interactions, while as $\alpha \to \infty$, the RQ kernel converges to the standard RBF kernel. The RQ kernel is robust to outliers and provides smooth transitions in feature representations, making it suitable for modeling data with multi-scale variations.

## A.2 THEORETICAL ANALYSIS

In this section, we provide our theoretical analysis for HP-GP. We first provide the explanation on the calculation details of intermediate variables referred. Afterwords, we provide the RKHS norm bound of HP-GP hyper-network outputs, before moving forward to conclusion of the Rademacher Complexity analysis and sampling bias within the Barron Space framework.

**Proposition 1** (Maximum Entropy Distribution). *Let $\mathcal{P}$ be the family of all probability density functions $p(\boldsymbol{\omega})$ over $\boldsymbol{\Omega}$ that satisfy the following moment constraints:*

*1. First-order moment: $\mathbb{E}_p[\boldsymbol{\omega}] = \boldsymbol{\mu}$, where $\boldsymbol{\mu} = \hat{E}_{x \sim \mathcal{D}} \boldsymbol{\mu}(x)$,*

*2. Second-order moment: $\mathbb{E}_p\left[(\boldsymbol{\omega} - \boldsymbol{\mu})^\top(\boldsymbol{\omega} - \boldsymbol{\mu})\right] = \boldsymbol{\Sigma}$, where*

$$\boldsymbol{\Sigma} = \hat{E}_{x \sim \mathcal{D}}\left[\boldsymbol{\Sigma}(x) + (\boldsymbol{\mu}(x) - \boldsymbol{\mu})^\top(\boldsymbol{\mu}(x) - \boldsymbol{\mu})\right].$$

*Then, the Gaussian distribution $q^*(\boldsymbol{\omega}) = \mathcal{N}(\boldsymbol{\omega}|\boldsymbol{\mu}, \boldsymbol{\Sigma})$ is the unique solution to the maximum entropy problem:*

$$\max_{p \in \mathcal{P}} - \int p(\boldsymbol{\omega}) \ln p(\boldsymbol{\omega}) d\boldsymbol{\omega}.$$

*Proof.* We prove this via Lagrangian optimization. Define the Lagrangian functional:

$$\mathcal{L}(p, \lambda, \Lambda) = -\int p(\boldsymbol{\omega}) \ln p(\boldsymbol{\omega}) d\boldsymbol{\omega} + \lambda^\top \left( \int \boldsymbol{\omega} p(\boldsymbol{\omega}) d\boldsymbol{\omega} - \boldsymbol{\mu} \right) + \mathrm{Tr}\left( \Lambda^\top \left( \int (\boldsymbol{\omega} - \boldsymbol{\mu})(\boldsymbol{\omega} - \boldsymbol{\mu})^\top p(\boldsymbol{\omega}) d\boldsymbol{\omega} - \boldsymbol{\Sigma} \right) \right),$$

where $\lambda \in \mathbb{R}^d$ and $\Lambda \in \mathbb{R}^{d \times d}$ (symmetric) are Lagrange multipliers.

Taking the functional derivative with respect to $p(\boldsymbol{\omega})$ and setting it to zero:

$$\frac{\delta \mathcal{L}}{\delta p(\boldsymbol{\omega})} = -\ln p(\boldsymbol{\omega}) - 1 + \lambda^\top \boldsymbol{\omega} + \mathrm{Tr}\left( \Lambda^\top (\boldsymbol{\omega} - \boldsymbol{\mu})(\boldsymbol{\omega} - \boldsymbol{\mu})^\top \right) = 0.$$

Rearranging gives:

$$p(\boldsymbol{\omega}) \propto \exp\left( \lambda^\top \boldsymbol{\omega} + (\boldsymbol{\omega} - \boldsymbol{\mu})^\top \Lambda (\boldsymbol{\omega} - \boldsymbol{\mu}) \right).$$

To match this to a Gaussian form, let $\Lambda = -\frac{1}{2}\boldsymbol{\Sigma}^{-1}$ and $\lambda = \boldsymbol{\Sigma}^{-1}\boldsymbol{\mu}$. Substituting these:

$$p(\boldsymbol{\omega}) \propto \exp\left( -\frac{1}{2}(\boldsymbol{\omega} - \boldsymbol{\mu})^\top \boldsymbol{\Sigma}^{-1} (\boldsymbol{\omega} - \boldsymbol{\mu}) \right),$$

which is the density of $\mathcal{N}(\boldsymbol{\omega} \mid \boldsymbol{\mu}, \boldsymbol{\Sigma})$.

For uniqueness, suppose another distribution $p(\boldsymbol{\omega}) \neq q^*(\boldsymbol{\omega})$ satisfies the same moment constraints. Since the Gaussian maximizes entropy, $H(p) < H(q^*)$, contradicting optimality. $\square$

**Proposition 2** (KL divergence conversion). *Equivalence of KL divergence constraints:* $\arg\min_{\boldsymbol{u}} \mathrm{KL}(q(\boldsymbol{\theta}_t \mid \mathcal{D}_t) \| p(\boldsymbol{\theta}_t)) = \arg\min_{\boldsymbol{u}} \mathrm{KL}(q(\boldsymbol{u}) \| p(\boldsymbol{u}; \boldsymbol{z}))$.

*Proof.* We can expand the KL divergence from $q(\boldsymbol{\theta}_t \mid \mathcal{D}_t)$ to $p(\boldsymbol{\theta}_t)$ as follows:

$$\mathrm{KL}(q(\boldsymbol{\theta}_t \mid \mathcal{D}_t) \| p(\boldsymbol{\theta}_t)) \tag{11}$$

$$= \mathbb{E}_{q(\boldsymbol{\theta}_t \mid \mathcal{D}_t)}(\log q(\boldsymbol{\theta}_t \mid \mathcal{D}_t) - \log p(\boldsymbol{\theta}_t))$$

$$= \mathbb{E}_{q(\boldsymbol{\theta}_t \mid \mathcal{D}_t)} \left( \log q(\boldsymbol{\theta}_t \mid \mathcal{D}_t) - \log p(\boldsymbol{F}^{-1}(\boldsymbol{\theta}_t)) - \log |\det(\frac{\partial \boldsymbol{F}^{-1}}{\partial \boldsymbol{\theta}_t})| \right)$$

$$= \int \left[ \log q(\boldsymbol{\theta}_t \mid \mathcal{D}_t) - \log p(\boldsymbol{F}^{-1}(\boldsymbol{\theta}_t)) - \log |\det(\frac{\partial \boldsymbol{F}^{-1}}{\partial \boldsymbol{\theta}_t})| \right] q(\boldsymbol{\theta}_t \mid \mathcal{D}_t) d\boldsymbol{\theta}_t.$$

With replacement $\boldsymbol{\theta}_t = \boldsymbol{F}(\boldsymbol{\omega}_t)$, we have:

$$\mathrm{KL}(q(\boldsymbol{\theta}_t \mid \mathcal{D}_t) \| p(\boldsymbol{\theta}_t)) \tag{12}$$

$$= \int \left[ \log q(\boldsymbol{F}(\boldsymbol{\omega}_t) \mid \mathcal{D}_t, \boldsymbol{z}) - \log p(\boldsymbol{\omega}_t \mid \boldsymbol{u}; \mathcal{D}_t, \boldsymbol{z}) + \log |\det(\frac{\partial \boldsymbol{F}}{\partial \boldsymbol{\omega}_t})| \right] q(\boldsymbol{\omega}_t \mid \mathcal{D}_t, \boldsymbol{z}) d\boldsymbol{\theta}_t$$

$$= \mathbb{E}_{q(\boldsymbol{\omega}_t \mid \mathcal{D}_t, \boldsymbol{z})}(\log q(\boldsymbol{\omega}_t \mid \mathcal{D}_t, \boldsymbol{z}) - \log p(\boldsymbol{\omega}_t \mid \boldsymbol{u}; \mathcal{D}_t, \boldsymbol{z}))$$

$$= \mathrm{KL}(q(\boldsymbol{\omega}_t \mid \mathcal{D}_t, \boldsymbol{z}) \| p(\boldsymbol{\omega}_t \mid \boldsymbol{u}; \mathcal{D}_t, \boldsymbol{z})).$$

This regularization can be further factorized, since both terms admit Gaussian distribution:

$$\mathrm{KL}(q(\boldsymbol{\omega}_t \mid \mathcal{D}_t, \boldsymbol{z}) \| p(\boldsymbol{\omega}_t \mid \boldsymbol{u}; \mathcal{D}_t, \boldsymbol{z})) \tag{13}$$

$$= \mathrm{KL}(\mathcal{N}(\boldsymbol{\omega}_t \mid \mathbb{E}_{\boldsymbol{x}_t \sim \mathcal{D}_t}[\boldsymbol{\mu}(\boldsymbol{x}_t)], \mathbb{E}_{\boldsymbol{x}_t \sim \mathcal{D}_t}[\boldsymbol{\Sigma}(\boldsymbol{x}_t) + (\boldsymbol{\mu}(\boldsymbol{x}_t) - \boldsymbol{\mu}_t)(\boldsymbol{\mu}(\boldsymbol{x}_t) - \boldsymbol{\mu}_t)^\top])$$

$$\| \mathcal{N}\left( \boldsymbol{\omega} \mid \boldsymbol{K}_{XZ}\boldsymbol{K}_{ZZ}^{-1}\boldsymbol{u}, \boldsymbol{K}_{XX} - \boldsymbol{K}_{XZ}\boldsymbol{K}_{ZZ}^{-1}\boldsymbol{K}_{ZX} \right))$$

$$= \frac{1}{2}\left[ \mathrm{tr}\left( (\boldsymbol{K}_{XX} - \boldsymbol{K}_{XZ}\boldsymbol{K}_{ZZ}^{-1}\boldsymbol{K}_{ZX})^{-1}(\boldsymbol{\Sigma}(\boldsymbol{x}_t) + (\boldsymbol{\mu}(\boldsymbol{x}_t) - \boldsymbol{\mu}_t)(\boldsymbol{\mu}(\boldsymbol{x}_t) - \boldsymbol{\mu}_t)^\top) \right) \right.$$

$$+ (\boldsymbol{K}_{XZ}\boldsymbol{K}_{ZZ}^{-1}\boldsymbol{u} - \boldsymbol{\mu}(\boldsymbol{x}_t))^\top (\boldsymbol{K}_{XX} - \boldsymbol{K}_{XZ}\boldsymbol{K}_{ZZ}^{-1}\boldsymbol{K}_{ZX})^{-1}(\boldsymbol{K}_{XZ}\boldsymbol{K}_{ZZ}^{-1}\boldsymbol{u} - \boldsymbol{\mu}(\boldsymbol{x}_t))$$

$$\left. - k + \ln\left( \frac{\det(\boldsymbol{K}_{XX} - \boldsymbol{K}_{XZ}\boldsymbol{K}_{ZZ}^{-1}\boldsymbol{K}_{ZX})}{\det(\boldsymbol{\Sigma}(\boldsymbol{x}_t) + (\boldsymbol{\mu}(\boldsymbol{x}_t) - \boldsymbol{\mu}_t)(\boldsymbol{\mu}(\boldsymbol{x}_t) - \boldsymbol{\mu}_t)^\top)} \right) \right]$$

According to the calculation on KL-divergence between multivariate Gaussian variables. The minimum of the regularization can be reached at

$$\mathbb{E}_{\boldsymbol{x}_t \sim \mathcal{D}_t}[\boldsymbol{\mu}(\boldsymbol{x}_t)] = \boldsymbol{K}_{XZ}\boldsymbol{K}_{ZZ}^{-1}\boldsymbol{u}, \quad \mathbb{E}_{\boldsymbol{x}_t \sim \mathcal{D}_t}[\boldsymbol{\Sigma}(\boldsymbol{x}_t)] = \boldsymbol{K}_{XX} - \boldsymbol{K}_{XZ}\boldsymbol{K}_{ZZ}^{-1}\boldsymbol{K}_{ZX},$$

meaning the posterior of $\boldsymbol{\omega}_t$ is expected to approach its prior. Meanwhile, the two terms can be connected with the inference procedure of Gaussian Process model:

$$q(\boldsymbol{\omega}_t \mid \mathcal{D}_t, \boldsymbol{z}) = \int p(\boldsymbol{\omega}_t \mid \boldsymbol{u}; \mathcal{D}_t, \boldsymbol{z}) q(\boldsymbol{u}) \, d\boldsymbol{u}. \tag{14}$$

Therefore, for a Gaussian Process to output posterior equal to its prior, the constraint is equivalent to $q(\boldsymbol{u}) = p(\boldsymbol{u}) = \mathcal{N}(\boldsymbol{u} \mid \boldsymbol{0}, \boldsymbol{K}_{ZZ})$, thus complete the deduction. $\qquad\square$

*Remark* 1. It is worth noting that although the regularization terms at two levels can be ensured with identical extreme value, the path of parameters during optimization are not guaranteed with similarity.

**Proposition 3** (Variables with tensor product under Gaussian distribution). *Let* $p(\boldsymbol{\omega}_{\mathrm{in}}) = \mathcal{N}(\boldsymbol{\omega}_{\mathrm{in}} \mid \boldsymbol{\mu}_{\mathrm{in}}, \boldsymbol{\Sigma}_{\mathrm{in}})$ *and* $p(\boldsymbol{\omega}_{\mathrm{out}}) = \mathcal{N}(\boldsymbol{\omega}_{\mathrm{out}} \mid \boldsymbol{\mu}_{\mathrm{out}}, \boldsymbol{\Sigma}_{\mathrm{out}})$ *be two independent random tensors. The tensor products* $\boldsymbol{\omega}_{\mathrm{in}} \otimes \boldsymbol{\omega}_{\mathrm{out}}$ *follows Gaussian distribution with the following parameters:*

$$p(\boldsymbol{\omega}_{\mathrm{in}} \otimes \boldsymbol{\omega}_{\mathrm{out}}) = \mathcal{N}\left(\boldsymbol{\omega}_{\mathrm{in}} \otimes \boldsymbol{\omega}_{\mathrm{out}} \mid \boldsymbol{\mu}_{\mathrm{in}} \otimes \boldsymbol{\mu}_{\mathrm{out}}, \boldsymbol{\Sigma}_{\mathrm{in}} \otimes [\boldsymbol{\mu}_{\mathrm{out}}]^2 + [\boldsymbol{\mu}_{\mathrm{in}}]^2 \otimes \boldsymbol{\Sigma}_{\mathrm{out}} + \boldsymbol{\Sigma}_{\mathrm{in}} \otimes \boldsymbol{\Sigma}_{\mathrm{out}}\right).$$

*Proof.* The expectation operator $\mathbb{E}$ is linear. Furthermore, since $\boldsymbol{\omega}_{\mathrm{in}}$ and $\boldsymbol{\omega}_{\mathrm{out}}$ are independent random vectors, the expectation of their tensor product factorizes into the tensor product of their expectations:

$$\mathbb{E}[\boldsymbol{\omega}_{\mathrm{in}} \otimes \boldsymbol{\omega}_{\mathrm{out}}] = \mathbb{E}[\boldsymbol{\omega}_{\mathrm{in}}] \otimes \mathbb{E}[\boldsymbol{\omega}_{\mathrm{out}}] = \boldsymbol{\mu}_{\mathrm{in}} \otimes \boldsymbol{\mu}_{\mathrm{out}}$$

The covariance matrix is defined as the expectation of the outer product of the deviation from the mean:

$$\mathrm{Cov}(\boldsymbol{\omega}_{\mathrm{in}} \otimes \boldsymbol{\omega}_{\mathrm{out}}) = \mathbb{E}\left[(\boldsymbol{\omega}_{\mathrm{in}} \otimes \boldsymbol{\omega}_{\mathrm{out}} - \boldsymbol{\mu}_{\mathrm{in}} \otimes \boldsymbol{\mu}_{\mathrm{out}}) \otimes (\boldsymbol{\omega}_{\mathrm{in}} \otimes \boldsymbol{\omega}_{\mathrm{out}} - \boldsymbol{\mu}_{\mathrm{in}} \otimes \boldsymbol{\mu}_{\mathrm{out}})\right]$$

Expand the terms based on the bilinear property of tensor products and regroup the terms into covariance on the single term, the computation of covariance term becomes:

$$\mathrm{Cov}(\boldsymbol{\omega}_{\mathrm{in}} \otimes \boldsymbol{\omega}_{\mathrm{out}}) = \boldsymbol{\Sigma}_{\mathrm{in}} \otimes \boldsymbol{\Sigma}_{\mathrm{out}} + \boldsymbol{\Sigma}_{\mathrm{in}} \otimes [\boldsymbol{\mu}_{\mathrm{out}}]^2 + [\boldsymbol{\mu}_{\mathrm{in}}]^2 \otimes \boldsymbol{\Sigma}_{\mathrm{out}}$$

On this basis, to prove that $\boldsymbol{\omega}_{\mathrm{in}} \otimes \boldsymbol{\omega}_{\mathrm{out}}$ follows a Gaussian distribution, we will show that any linear transformation of it yields a univariate Gaussian random variable.

Consider an arbitrary fixed matrix $\boldsymbol{A}$ of appropriate dimensions. We need to show that the random variable:

$$Z = \langle \boldsymbol{A}, \boldsymbol{\omega}_{\mathrm{in}} \otimes \boldsymbol{\omega}_{\mathrm{out}} \rangle$$

follows a Gaussian distribution, where $\langle \cdot, \cdot \rangle$ denotes the appropriate inner product.

Using the property of tensor products, we can express this inner product as:

$$Z = \langle \boldsymbol{A}, \boldsymbol{\omega}_{\mathrm{in}} \otimes \boldsymbol{\omega}_{\mathrm{out}} \rangle = \boldsymbol{\omega}_{\mathrm{in}}^\top \boldsymbol{A} \boldsymbol{\omega}_{\mathrm{out}}$$

Since $\boldsymbol{\omega}_{\mathrm{in}}$ and $\boldsymbol{\omega}_{\mathrm{out}}$ are independent Gaussian random vectors, their joint distribution is Gaussian. The quadratic form $\boldsymbol{\omega}_{\mathrm{in}}^\top \boldsymbol{A} \boldsymbol{\omega}_{\mathrm{out}}$ is a linear function of the elements of the joint random vector $(\boldsymbol{\omega}_{\mathrm{in}}, \boldsymbol{\omega}_{\mathrm{out}})$.

Specifically, we can write:

$$Z = \sum_{i,j} A_{ij} \omega_{\mathrm{in},i} \omega_{\mathrm{out},j}$$

where $\omega_{\mathrm{in},i}$ and $\omega_{\mathrm{out},j}$ are the components of $\boldsymbol{\omega}_{\mathrm{in}}$ and $\boldsymbol{\omega}_{\mathrm{out}}$ respectively.

This is a linear combination of the products of components of two independent Gaussian random vectors. Since the joint distribution of $(\boldsymbol{\omega}_{\mathrm{in}}, \boldsymbol{\omega}_{\mathrm{out}})$ is Gaussian, and linear transformations of Gaussian random vectors remain Gaussian, it follows that any linear functional of $\boldsymbol{\omega}_{\mathrm{in}} \otimes \boldsymbol{\omega}_{\mathrm{out}}$ is Gaussian. $\quad\square$

**Definition 1.** (RKHS Space, (Kanagawa et al., 2018)) A reproducing kernel Hilbert space (RKHS) space is a Hilbert space $\mathcal{H}_k$ of functions on $X$ equipped with an inner product $\langle \cdot, \cdot \rangle_{\mathcal{H}_k}$, satisfying

$$k(\cdot, x) \in \mathcal{H}_k \quad and \quad f(x) = \langle f, k(\cdot, x) \rangle_{\mathcal{H}_k}, \forall x \in \mathcal{X}, f \in \mathcal{H}_k, \tag{15}$$

where $k$ be a positive definite kernel on non empty set $\mathcal{X}$. This space is also endowed with RKHS norm defined as

$$\|f\|_{\mathcal{H}_k} := \left\| \sum_{i=1}^{\infty} c_i k(\cdot, x_i) \right\| = \sum_{i,j=1}^{\infty} c_i c_j k(x_i, x_j), c_i \in \mathbb{R}, x_i \in \mathcal{X}. \tag{16}$$

**Definition 2.** (Barron Space, (Zhang, 2023)) A function $f : \mathcal{X} \to \mathbb{R}$ belongs to the Barron space $\mathcal{B}$ if it admits the following representation:

$$f(x) = \int_{\Omega} \omega_{t,0}^{(2)} \sigma((\boldsymbol{\omega}_{t,0}^{(1)})^{\top} x + \omega_{t,1}^{(1)}) \rho(d\omega_{t,0}^{(1)}, d\boldsymbol{\omega}_{t,1}^{(1)}, d\omega_{t,0}^{(2)}), \forall x \in \mathcal{X}, \tag{17}$$

where $\Omega = \mathbb{R}^1 \times \mathbb{R}^d \times \mathbb{R}^1$, $\rho$ denotes a probability distribution on $(\Omega, \Sigma_\Omega)$ with $\Sigma_\Omega$ being a Borel $\sigma$-algebra on $\Omega$, $\sigma(z) = \max(z, 0)$ denotes the ReLU activation function. This space is also endowed with Barron norm defined as

$$\|f\|_{\mathcal{B}} := \inf_{\rho} \left\{ \max_{(\boldsymbol{\omega}_t^{(1)}, \omega_t^{(2)}) \in \text{supp}(\rho)} |\omega_t^{(2)}| \|\boldsymbol{\omega}_t^{(1)}\|_1 : f(x) = \int_{\Omega} \omega_{t,0}^{(2)} \sigma((\boldsymbol{\omega}_{t,0}^{(1)})^{\top} x + \omega_{t,1}^{(1)}) \rho(d\boldsymbol{\omega}_t^{(1)}, d\omega_t^{(2)}) \right\}. \tag{18}$$

*Remark* 2. Since the representation of a random function $f : \mathcal{X} \to \mathbb{R}$ in Barron Space is not guaranteed uniqueness, the Barron norm is defined as the infimum of $\max_{(\boldsymbol{\omega}_t^{(1)}, \omega_t^{(2)}) \in \text{supp}(\rho)} \left[ |\omega_{t,0}^{(2)}| (\|\boldsymbol{\omega}_{t,0}^{(1)}\|_1 + |\omega_{t,1}^{(1)}|) \right]$ across distributions. Yet under the scenario of standard NN with HP-GP, random distribution $\rho$ is replaced by the posterior distribution $q$, exempting the operation of inferior lower bound $\inf_\rho$.

**Theorem 1.** *Assuming that input domain $\mathcal{X}$ is compact and $k$ is continuous, the regularization of inducing variables is bounded by $KL[q(\boldsymbol{u}^{(d)})\|p(\boldsymbol{u}^{(d)}; \boldsymbol{z}^{(d)})] \leq B_{KL}, d \in \{in, out\}$, and the power of episode $|\mathcal{D}_t| < \infty$. With probability $\geq (1-\delta)^2$, the following assertion holds for HP-GP output based on bounded inputs $\|x_t\|_1 \leq 1$:*

$$\|\boldsymbol{\theta}_t\|_1 \leq 2D \max_j \sum_{j=1}^{D} |w_j|(1 + L_j)[\frac{d}{\sqrt{\delta}}(8C_2^2|\mathcal{D}_t|(\max_i \|\phi_i\|_{L_\infty(\mathcal{D}_t)})^2 B_{KL}$$

$$+ C_1 \max_i \|\phi_i\|_{L_\infty(\mathcal{D}_t)}^4 + C_0 B_{KL}) + 2C_2^2|\mathcal{D}_t| \max_i \|\phi_i\|_{L_\infty(\mathcal{D}_t)}^2 B_{KL}]^2,$$

*where $C_0, C_1, C_2$ are constants, $\dim(\boldsymbol{\omega}_t) = D$, $k(\boldsymbol{x}, \boldsymbol{x}) = \sum_{i=1}^{\infty} \lambda_i \phi_i(\boldsymbol{x})^2$ with assumption $\sum_{i=1}^{\infty} \lambda_i^p < \infty, \forall p \in (0, 1)$ $\boldsymbol{w}$ is the weight parameter, and spline function is assumed with Lipchitz continuity with Lipchitz constant $|Spline_j(x)| \leq L_j|x|$.*

*Proof.* Consider the prior and posterior distribution of inducing points for the two tensors $\boldsymbol{u}^{(d)}, d \in \{in, out\}$: $p(\boldsymbol{u}^{(d)}; \boldsymbol{z}^{(d)}) = \mathcal{N}(\boldsymbol{u}^{(d)} \mid \boldsymbol{0}, \boldsymbol{K}_{zz}^{(d)}), q(\boldsymbol{u}^{(d)}) = \mathcal{N}(\boldsymbol{u}^{(d)} \mid \boldsymbol{m}^{(d)}, \boldsymbol{S}^{(d)})$. At the starting point, we calculate the scope of $\boldsymbol{\mu}^{(d)}$ and $\boldsymbol{\Sigma}^{(d)}$ based on the KL divergence constraint $KL[q(\boldsymbol{u}^{(d)})\|p(\boldsymbol{u}^{(d)}; \boldsymbol{z}^{(d)})] \leq B_{KL}$, which can be expanded as:

$$\frac{1}{2}\Big[\text{tr}\big((\boldsymbol{K}_{zz}^{(d)})^{-1}\boldsymbol{S}^{(d)}\big) + (\boldsymbol{m}^{(d)})^{\top}(\boldsymbol{K}_{zz}^{(d)})^{-1}\boldsymbol{m}^{(d)} - |d| + \ln\det(\boldsymbol{K}_{zz}^{(d)}) - \ln\det(\boldsymbol{S}^{(d)})\Big] \leq B_{KL}. \tag{19}$$

For $(\boldsymbol{K}_{zz}^{(d)})^{-1}\boldsymbol{S}^{(d)} \succ 0$, $\text{tr}\big((\boldsymbol{K}_{zz}^{(d)})^{-1}\boldsymbol{S}^{(d)}\big) + \ln\det(\boldsymbol{K}_{zz}^{(d)}) - \ln\det(\boldsymbol{S}^{(d)}) \geq |d|$. The minimum occurs at $\boldsymbol{S}^{(d)} = \boldsymbol{K}_{zz}^{(d)}$ (Boucheron et al., 2013), yielding:

$$\|\boldsymbol{\mu}^{(d)}\|_{\mathcal{H}_k}^2 = (\boldsymbol{m}^{(d)})^{\top}(\boldsymbol{K}_{zz}^{(d)})^{-1}\boldsymbol{m}^{(d)} = 2KL[q(\boldsymbol{u}^{(d)})\|p(\boldsymbol{u}^{(d)}; \boldsymbol{z}^{(d)})] \leq 2B_{KL}. \tag{20}$$

According to the reproducing property, we acquire $tr((\alpha^{(d)})^\top(\boldsymbol{K}_{zz}^{(d)} - \boldsymbol{S}^{(d)})\alpha^{(d)}) \leq C_0^{(d)} B_{KL}$, and $tr(k_{xx}^{(d)}) \leq C_1^{(d)} \max_i \|\phi_i^{(d)}\|_{L_\infty(\mathcal{D}_t)}^4$.

To note that $\|\phi_i^{(d)}\|_{L_\infty(\mathcal{D}_t)} < \infty, \forall i$ is guaranteed by the boundedness theorem, since input domain $\mathcal{X}$ is compact and $k$ is continuous.

With both term combined, we arrive at the bound for the trace of $\boldsymbol{\Sigma}^{(d)}(x_t)$ with

$$tr(\boldsymbol{\Sigma}^{(d)}(x_t)) = tr(k_{xx}^{(d)} - tr((\boldsymbol{\alpha}^{(d)})^\top(\boldsymbol{K}_{zz}^{(d)} - \boldsymbol{S}^{(d)})\boldsymbol{\alpha}^{(d)})) \leq C_1^{(d)} \max_i \|\phi_i^{(d)}\|_{L_\infty(\mathcal{D}_t)}^4 + C_0^{(d)} B_{KL}, \tag{21}$$

where $C_0^{(d)}$ and $C_1^{(d)}$ are constants.

Assuming that input domain $\mathcal{X}$ is compact and $k$ is continuous, the following transformation converts the scope of RKHS norm into bound on $L_p$ norm,

$$\|\boldsymbol{\mu}^{(d)}\|_{L_p(\mathcal{D}_t)}^p = \int_{\mathcal{D}_t} |\boldsymbol{\mu}^{(d)}(\boldsymbol{x})|^p d\boldsymbol{x} = \int_{\mathcal{D}_t} |\langle \mu^{(d)}, k^{(d)}(\boldsymbol{x},\cdot)\rangle_{\mathcal{H}_k}|^p d\boldsymbol{x} \leq \|\boldsymbol{\mu}^{(d)}\|_{\mathcal{H}_k}^p \int_{\mathcal{D}_t} k^{(d)}(\boldsymbol{x},\boldsymbol{x})^{p/2} d\boldsymbol{x}, \tag{22}$$

where the second and the third operation follows from the reproducing property of RKHS Space and Cauchy-Schwarz inequality respectively.

For the condition of finite norms, taking the $1/p$-th root gives,

$$\|\boldsymbol{\mu}^{(d)}\|_{L_p(\mathcal{D}_t)} \leq \left(\int_{\mathcal{D}_t} k^{(d)}(\boldsymbol{x},\boldsymbol{x})^{p/2} d\boldsymbol{x}\right)^{1/p} \|\boldsymbol{\mu}^{(d)}\|_{\mathcal{H}_k} \tag{23}$$

$$\leq \left(\int_{\mathcal{D}_t} (\sum_{i=1}^\infty \lambda_i^{(d)} \phi_i^{(d)}(x)^2)^{p/2} d\boldsymbol{x}\right)^{1/p} \sqrt{2B_{\text{KL}}}.$$

For $p \geq 2$, according to Jenson inequality,

$$\int_{\mathcal{D}_t} (\sum_{i=1}^\infty \lambda_i^{(d)} \phi_i^{(d)}(x)^2)^{p/2} d\boldsymbol{x} \leq (\sum_{i=1}^\infty (\lambda_i^{(d)})^{p/2}) \int_{\mathcal{D}_t} \phi_i^{(d)}(x)^p d\boldsymbol{x} \tag{24}$$

$$\leq |\mathcal{D}_t| \sum_{i=1}^\infty (\lambda_i^{(d)})^{p/2} \|\phi_i^{(d)}\|_{L_\infty(\mathcal{D}_t)}^p \leq C_2^{(d)} |\mathcal{D}_t| \max_i \|\phi_i^{(d)}\|_{L_\infty(\mathcal{D}_t)}^p$$

For $1 \leq p < 2$, according to Holder inequality,

$$\int_{\mathcal{D}_t} (\sum_{i=1}^\infty \lambda_i^{(d)} \phi_i^{(d)}(x)^2)^{p/2} d\boldsymbol{x} \leq (\sum_{i=1}^\infty \lambda_i^{(d)})^{1/q} (\sum_{i=1}^\infty (\lambda_i^{(d)})^{p/2}) \int_{\mathcal{D}_t} (\phi_i(x)^{(d)})^p d\boldsymbol{x} \tag{25}$$

$$\leq |\mathcal{D}_t| (\sum_{i=1}^\infty \lambda_i^{(d)})^{1/q} (\sum_{i=1}^\infty (\lambda_i^{(d)})^{p/2}) \|\phi_i^{(d)}\|_{L_\infty(\mathcal{D}_t)}^p \leq C_2^{(d)} |\mathcal{D}_t| \max_i \|\phi_i^{(d)}\|_{L_\infty(\mathcal{D}_t)}^p$$

where $\frac{p}{2} + \frac{1}{q} = 1$, and $C_2$ is a constant.

The function norm naturally implies the scope of variable norm via its definition

$$\|\boldsymbol{\mu}^{(d)}(x_t)\|_p = \|\boldsymbol{\mu}^{(d)}\|_{L_p(\mathcal{D}_t)} \leq C_2^{(d)} |\mathcal{D}_t| \max_i \|\phi_i^{(d)}\|_{L_\infty(\mathcal{D}_t)}^p \sqrt{2B_{\text{KL}}}, \forall \|x_t\|_p \leq 1. \tag{26}$$

A similar deduction provides the bound under the $L_\infty(\mathcal{D}_t)$ norm as follows:

$$\|\boldsymbol{\mu}^{(d)}(x_t)\|_\infty = \|\boldsymbol{\mu}^{(d)}\|_{L_\infty(\mathcal{D}_t)} \leq \|\boldsymbol{\mu}^{(d)}\|_{\mathcal{H}_k} \sup_{\boldsymbol{x}\in\mathcal{D}_t} \sqrt{k^{(d)}(\boldsymbol{x},\boldsymbol{x})} \leq \|\boldsymbol{\mu}^{(d)}\|_{\mathcal{H}_k} \sup_{\boldsymbol{x}\in\mathcal{D}_t} \sqrt{\sum_{i=1}^\infty \lambda_i^{(d)} \phi_i^{(d)}(x)^2}$$

$$\tag{27}$$

$$\leq \max_i \|\phi_i^{(d)}\|_{L_\infty(\mathcal{D}_t)} \|\boldsymbol{\mu}^{(d)}\|_{\mathcal{H}_k} \sup_{\boldsymbol{x}\in\mathcal{D}_t} \sqrt{\sum_{i=1}^\infty \lambda_i^{(d)}} \leq C_2^{(d)} \max_i \|\phi_i^{(d)}\|_{L_\infty(\mathcal{D}_t)} \sqrt{2B_{KL}}.$$

According to the norm factorization of tensor product, we acquire the 1-norm bound for eventual mean output $\|\boldsymbol{\mu}(x_t)\|_1 = \|\boldsymbol{\mu}^{\text{in}}(x_t)\|_1\|\boldsymbol{\mu}^{\text{out}}(x_t)\|_\infty \leq 2C_2^2|\mathcal{D}_t|\max_i\|\phi_i\|_{L_\infty(\mathcal{D}_t)}^2 B_{\text{KL}}$.

And the aggregation on covariance matrix is implemented by $\boldsymbol{\Sigma}(\boldsymbol{x}_t) = \boldsymbol{\Sigma}_{\text{in}}(\boldsymbol{x}_t) \otimes [\boldsymbol{\mu}_{\text{out}}(\boldsymbol{x}_t)]^2 + [\boldsymbol{\mu}_{\text{in}}(\boldsymbol{x}_t)]^2 \otimes \boldsymbol{\Sigma}_{\text{out}}(\boldsymbol{x}_t) + \boldsymbol{\Sigma}_{\text{in}}(\boldsymbol{x}_t) \otimes \boldsymbol{\Sigma}_{\text{out}}(\boldsymbol{x}_t)$, which provides the following RKHS bound for aggregated covariance matrx:

$$tr(\boldsymbol{\Sigma}(x_t)) \leq tr(\boldsymbol{\Sigma}_{\text{in}}(x_t))tr([\boldsymbol{\mu}_{\text{out}}(\boldsymbol{x}_t)]^2) + tr(\boldsymbol{\Sigma}_{\text{out}}(x_t))tr([\boldsymbol{\mu}_{\text{in}}(\boldsymbol{x}_t)]^2) + tr(\boldsymbol{\Sigma}_{\text{in}}(x_t))tr(\boldsymbol{\Sigma}_{\text{out}}(x_t))$$

$$(28)$$

$$\leq [4C_2^2|\mathcal{D}_t|(\max_i\|\phi_i\|_{L_\infty(\mathcal{D}_t)})^2 B_{\text{KL}} + C_1\max_i\|\phi_i\|_{L_\infty(\mathcal{D}_t)}^4 + C_0 B_{KL}][C_1\max_i\|\phi_i\|_{L_\infty(\mathcal{D}_t)}^4 + C_0 B_{KL}]$$

$$\leq [4C_2^2|\mathcal{D}_t|(\max_i\|\phi_i\|_{L_\infty(\mathcal{D}_t)})^2 B_{\text{KL}} + C_1\max_i\|\phi_i\|_{L_\infty(\mathcal{D}_t)}^4 + C_0 B_{KL}]^2, \qquad (29)$$

where $C_i = \max\{C_i^{\text{out}}, C_i^{\text{in}}\}, i \in \{0,1\}, \|\phi_i\|_{L_\infty(\mathcal{D}_t)} = \max\{\|\phi_i^{\text{out}}\|_{L_\infty(\mathcal{D}_t)}, \|\phi_i^{\text{in}}\|_{L_\infty(\mathcal{D}_t)}\}$.

According to the multivariate Chebyshev inequality:

$$\mathbb{P}\left(\|\boldsymbol{\mu}(x_t) - \underset{x_t\sim\mathcal{D}_t}{\boldsymbol{E}}\boldsymbol{\mu}(x_t)\|_1 \geq k\right) \leq \frac{d^2 \text{tr}\left(\underset{x_t\sim\mathcal{D}_t}{\boldsymbol{Var}}\boldsymbol{\mu}(x_t)\right)}{k^2}, \forall k > 0. \qquad (30)$$

Equivalently, with probability$\geq 1 - \delta$,

$$\|\boldsymbol{\mu}(x_t) - \underset{x_t\sim\mathcal{D}_t}{\boldsymbol{E}}\boldsymbol{\mu}(x_t)\|_1 \leq d\sqrt{\frac{\text{tr}\left(\underset{x_t\sim\mathcal{D}_t}{\boldsymbol{Var}}\boldsymbol{\mu}(x_t)\right)}{\delta}} \qquad (31)$$

$$\leq d\sqrt{\frac{\left(\underset{x_t\sim\mathcal{D}_t}{\boldsymbol{E}}\|\boldsymbol{\mu}\|_2^2(x_t)\right)}{\delta}} \leq 2dC_2^2|\mathcal{D}_t|\max_i\|\phi_i^{\text{in}}\|_{L_\infty(\mathcal{D}_t)}^2 B_{\text{KL}}/\sqrt{\delta}.$$

Likewise, with probability$\geq 1 - \delta$, the following equation holds,

$$\|\boldsymbol{\omega}_t - \underset{x_t\sim\mathcal{D}_t}{\boldsymbol{E}}\boldsymbol{\mu}(x_t)\|_1 \leq d\sqrt{\frac{\text{tr}\left(\underset{x_t\sim\mathcal{D}_t}{\boldsymbol{E}}\boldsymbol{\Sigma}(x_t) + \underset{x_t\sim\mathcal{D}_t}{\boldsymbol{Var}}\boldsymbol{\mu}(x_t)\right)}{\delta}} \leq d\sqrt{\frac{\text{tr}\left(\underset{x_t\sim\mathcal{D}_t}{\boldsymbol{E}}\boldsymbol{\Sigma}(x_t)\right)}{\delta}} + d\sqrt{\frac{\text{tr}\left(\underset{x_t\sim\mathcal{D}_t}{\boldsymbol{Var}}\boldsymbol{\mu}(x_t)\right)}{\delta}}$$

$$(32)$$

$$\leq \frac{d}{\sqrt{\delta}}\left(6C_2^2|\mathcal{D}_t|(\max_i\|\phi_i\|_{L_\infty(\mathcal{D}_t)})^2 B_{\text{KL}} + C_1\max_i\|\phi_i\|_{L_\infty(\mathcal{D}_t)}^4 + C_0 B_{KL}\right).$$

With both equation combined, with probability $\geq (1 - \delta)^2$, the following assertion holds,

$$\|\boldsymbol{\omega}_t\|_1 \leq \|\boldsymbol{\omega}_t - \underset{x_t\sim\mathcal{D}_t}{\boldsymbol{E}}\boldsymbol{\mu}(x_t)\|_1 + \|\boldsymbol{\mu}(x_t) - \underset{x_t\sim\mathcal{D}_t}{\boldsymbol{E}}\boldsymbol{\mu}(x_t)\|_1 + \|\boldsymbol{\mu}(x_t)\|_1 \qquad (33)$$

$$\leq \frac{d}{\sqrt{\delta}}\left(8C_2^2|\mathcal{D}_t|(\max_i\|\phi_i\|_{L_\infty(\mathcal{D}_t)})^2 B_{\text{KL}} + C_1\max_i\|\phi_i\|_{L_\infty(\mathcal{D}_t)}^4 + C_0 B_{KL}\right)$$

$$+ 2C_2^2|\mathcal{D}_t|\max_i\|\phi_i\|_{L_\infty(\mathcal{D}_t)}^2 B_{\text{KL}}.$$

With the bound on intermediate variable $\boldsymbol{\omega}_t$, we can proceed with the calculation of Inverse Autoregressive Flow processing procedure. The general procedure of the dimension-wise calculation can be formulated as: $\theta_t^i = \omega_t^i\exp\alpha_t^i + \beta_t^i, \beta_t^i = f_{\beta^i}(\boldsymbol{\omega}_t^{1:i-1}), \alpha_t^i = f_{\alpha^i}(\boldsymbol{\omega}_t^{1:i-1})$. Under our experimental setting, $f_{\beta^i}$ and $f_{\alpha^i}$ are realized with KAN with default configurations, rendering the following transformation: $f(\boldsymbol{\omega}) = W(\text{SiLU}(\omega) + \text{Spline}(\omega))$.

A natural bound becomes

$$\|f(\boldsymbol{\theta}_t^{1:i-1})\|_1 \leq \sum_{j=1}^{i-1}|w_j|(|\boldsymbol{\omega}_t^i| + |Spline_j(\omega_t^j)|) \leq \sum_{j=1}^{i-1}|w_j||\omega_t^j|(1+L_j) \leq \|\boldsymbol{\omega}_t\|_1\max_j\sum_{j=1}^{D}|w_j|(1+L_j),$$

$$(34)$$

And the eventual output of the IAF layer can be calculated by iteratively applying triangle inequality as follows:

$$\|\boldsymbol{\theta}_t^{1:D}\|_1 \leq \|\boldsymbol{\theta}_t^{1:D-1}\|_1 + |\omega_t^D||\exp\alpha_t^D| + |\beta_t^D| \tag{35}$$

$$\leq \sum_{i=1}^{D} |\omega_t^i||\exp\alpha_t^i| + |\beta_t^i| \leq 2\|\boldsymbol{\omega}_t\|_1^2 D \max_j \sum_{j=1}^{D} |w_j|(1+L_j)$$

$$= 2D \max_j \sum_{j=1}^{D} |w_j|(1+L_j)[\frac{d}{\sqrt{\delta}}(8C_2^2|\mathcal{D}_t|(\max_i \|\phi_i\|_{L_\infty(\mathcal{D}_t)})^2 B_{\mathrm{KL}}$$

$$+ C_1 \max_i \|\phi_i\|_{L_\infty(\mathcal{D}_t)}^4 + C_0 B_{KL}) + 2C_2^2|\mathcal{D}_t| \max_i \|\phi_i\|_{L_\infty(\mathcal{D}_t)}^2 B_{\mathrm{KL}}]^2.$$

$\square$

**Theorem 2.** *(Rademacher complexity bound) With all assumptions in Theorem 1 hold, for the function class $\mathcal{F}_Q = \{f_{\boldsymbol{\omega}_t} \in \mathcal{B}\}$, and external output weight$\omega_{t,0}^{(2)}$, with probability $\geq (1-\delta)^2$, the empirical Rademacher complexity satisfies:*

$$\widehat{\mathfrak{R}}_n(\mathcal{F}_Q) \leq 4|\omega_{t,0}^{(2)}|D\max_j \sum_{j=1}^{D} |w_j|(1+L_j)[\frac{d}{\sqrt{\delta}}(8C_2^2|\mathcal{D}_t|(\max_i \|\phi_i\|_{L_\infty(\mathcal{D}_t)})^2 B_{KL}$$

$$+ C_1 \max_i \|\phi_i\|_{L_\infty(\mathcal{D}_t)}^4 + C_0 B_{KL}) + 2C_2^2|\mathcal{D}_t| \max_i \|\phi_i\|_{L_\infty(\mathcal{D}_t)}^2 B_{\mathrm{KL}}]^2 \sqrt{\frac{2\ln(2d)}{|\mathcal{D}_t|}}.$$

*Proof.* With probability $\geq (1-\delta)^2$, the Barron norm of the network considered can be bounded by,

$$\|f\|_{\mathcal{B}} = \max_{(\boldsymbol{\omega}_t^{(1)}, \boldsymbol{\omega}_t^{(2)}) \in \mathrm{supp}(q)} \left[ |\omega_{t,0}^{(2)}|(\|\boldsymbol{\omega}_{t,0}^{(1)}\|_1 + |\omega_{t,1}^{(1)}|) \right] \tag{36}$$

$$\leq \max_{(\boldsymbol{\omega}_t^{(1)}, \boldsymbol{\omega}_t^{(2)}) \in \mathrm{supp}(q)} \left[ |\omega_{t,0}^{(2)}|\|\boldsymbol{\omega}_t^{(1)}\|_1 \right]$$

$$\leq 2|\omega_{t,0}^{(2)}|D\max_j \sum_{j=1}^{D} |w_j|(1+L_j)[\frac{d}{\sqrt{\delta}}(8C_2^2|\mathcal{D}_t|(\max_i \|\phi_i\|_{L_\infty(\mathcal{D}_t)})^2 B_{\mathrm{KL}}$$

$$+ C_1 \max_i \|\phi_i\|_{L_\infty(\mathcal{D}_t)}^4 + C_0 B_{KL}) + 2C_2^2|\mathcal{D}_t| \max_i \|\phi_i\|_{L_\infty(\mathcal{D}_t)}^2 B_{\mathrm{KL}}]^2,$$

where the second inequality derives from Theorem 1.

According to Theorem 3 in E et al. (2021), the Rademacher complexity of function class $\mathcal{F}_Q = \{f \in \mathcal{B} : \|f\|_{\mathcal{B}} \leq Q\}$ on data set sized n: $S = \{\boldsymbol{x}_1, \boldsymbol{x}_2, \ldots, \boldsymbol{x}_n\}$is bounded by $\widehat{\mathfrak{R}}_n(\mathcal{F}_Q) \leq 2Q\sqrt{\frac{2\ln(2d)}{n}}$.

Consequently, with probability $\geq (1-\delta)^2$, the Rademacher complexity of our function considered satisfies,

$$\widehat{\mathfrak{R}}_n(\mathcal{F}_Q) \leq 4|\omega_{t,0}^{(2)}|D\max_j \sum_{j=1}^{D} |w_j|(1+L_j)[\frac{d}{\sqrt{\delta}}(8C_2^2|\mathcal{D}_t|(\max_i \|\phi_i\|_{L_\infty(\mathcal{D}_t)})^2 B_{\mathrm{KL}} \tag{37}$$

$$+ C_1 \max_i \|\phi_i\|_{L_\infty(\mathcal{D}_t)}^4 + C_0 B_{KL}) + 2C_2^2|\mathcal{D}_t| \max_i \|\phi_i\|_{L_\infty(\mathcal{D}_t)}^2 B_{\mathrm{KL}}]^2 \sqrt{\frac{2\ln(2d)}{|\mathcal{D}_t|}}.$$

Thus complete the proof. $\square$

**Theorem 3.** *(Approximation Error) With all assumptions in Theorem 1 hold, for the function class $\mathcal{F}_Q = \{f_{\boldsymbol{\omega}_t} \in \mathcal{B}\}$, with probability $\geq (1-\delta)^2$, there exists a standard neural network$f_{m,\boldsymbol{\Omega}}(\boldsymbol{x}) = \frac{1}{m}\sum_{k=1}^{m} \boldsymbol{\Omega}_{k,0}^{(2)}\sigma((\boldsymbol{\Omega}_{k,0}^{(1)})^\top \boldsymbol{x} + \boldsymbol{\Omega}_{k,1}^{(2)})$ that satisfies*

$$\|f(\cdot) - f_{m,\boldsymbol{\Omega}}(\cdot)\|^2 \leq \frac{12}{m}|\omega_{t,0}^{(2)}|^2 D^2 [\max_j \sum_{j=1}^{D} |w_j|(1+L_j)]^2 [\frac{d}{\sqrt{\delta}}(8C_2^2|\mathcal{D}_t|(\max_i \|\phi_i\|_{L_\infty(\mathcal{D}_t)})^2 B_{KL}$$

$$+ C_1 \max_i \|\phi_i\|_{L_\infty(\mathcal{D}_t)}^4 + C_0 B_{KL}) + 2C_2^2|\mathcal{D}_t| \max_i \|\phi_i\|_{L_\infty(\mathcal{D}_t)}^2 B_{KL}]^4.$$

*Proof.* According to Theorem 1 in E et al. (2021), For any $f \in \mathcal{B}$ and $m > 0$, there exists a standard neural network $f_{m,\Omega}(x) = \frac{1}{m}\sum_{k=1}^{m}\Omega_{k,0}^{(2)}\sigma((\Omega_{k,0}^{(1)})^{\top}x + \Omega_{k,1}^{(2)})$, such that $\|f(\cdot) - f_m(\cdot;\Omega)\|^2 \leq \frac{3\|f\|_{\mathcal{B}}^2}{m}$,

Meanwhile,

$$\|f\|_{\mathcal{B}} \leq 2|\omega_{t,0}{}^{(2)}|D\max_{j}\sum_{j=1}^{D}|w_j|(1+L_j)[\frac{d}{\sqrt{\delta}}(8C_2^2|\mathcal{D}_t|(\max_{i}\|\phi_i\|_{L_\infty(\mathcal{D}_t)})^2 B_{\text{KL}} \tag{38}$$

$$+ C_1\max_{i}\|\phi_i\|_{L_\infty(\mathcal{D}_t)}^4 + C_0 B_{KL}) + 2C_2^2|\mathcal{D}_t|\max_{i}\|\phi_i\|_{L_\infty(\mathcal{D}_t)}^2 B_{\text{KL}}]^2.$$

Thus complete the proof. $\square$

### A.3 Training Details

A formatting description on the general implementation procedure of HP-GP is provided in Algorithm 1.

---

**Algorithm 1** Meta-training and meta-testing procedure of Hierarchical Parametrization with Gaussian Process in a Few-Shot Learning Setting.

---

**Require:** Training set $\mathcal{D}_{\text{train}} = \{\mathcal{D}_t\}_{t=1}^{T}$ and test set $\mathcal{D}_{\text{test}} = (\mathcal{D}_S^*, \mathcal{D}_Q^*)$.
**Require:** Variational inducing variable $z, u$ and normalizing flow latent variables $w$.
**Require:** Learning rate for training $\eta_{\text{train}}$.
  **function** TRAIN($\mathcal{D}_{\text{train}}, z, u, w, \eta_{\text{train}}$)
  **repeat**
    Sample episode from training set $\mathcal{D}_t = (\mathcal{D}_S^t, \mathcal{D}_Q^t) \sim \mathcal{D}_{\text{train}}$.
    Update $(z, u, w) \leftarrow (z, u, w) - \eta_{\text{train}}\nabla_{z,u,w}\mathcal{L}(\mathcal{D}_t)$.          ▷ Equation 6
  **until** Converge
  **return** $\hat{z}, \hat{u}, \hat{w}$.

  **function** TEST($\mathcal{D}_{\text{test}}, \hat{z}, \hat{u}, \hat{w}$)
  Compute parameter $q(\hat{\theta} \mid x_S, x_Q), x_S \sim \mathcal{D}_S^*, x_Q \sim \mathcal{D}_Q^*$.
  Calculate output $y_Q \sim p(y_Q \mid x_Q, \hat{\theta}), x_Q \sim \mathcal{D}_Q^*$.
  **return** $y_Q$.

---

#### A.3.1 Time Series Prediction

The NASDAQ dataset[1] consists of stock price data for 104 companies listed under the NASDAQ 100 index, along with the index value itself. The data is sampled at a one-minute frequency over a period of 191 trading days, from July 26, 2016, to April 28, 2017. Each trading day contains 391 data points for individual companies and 390 data points for the NASDAQ 100 index, with alignment such that data points 2 through 391 of the companies correspond to data points 1 through 390 of the index.

For this study, we utilize the NASDAQ dataset with padding, formatting the data into a consistent structure with 390 points per day over a 105-day interval. To construct meta-training tasks, we use the first 70% of the data from the NASDAQ 100 index, while meta-test episodes for the in-range scenario are derived from the final 30% of the data. Additionally, out-of-range inference episodes are created by using the entire stock price time series from the dataset. In each episode, we randomly sample 10 consecutive data points from the time series, dividing them into 5 support points and 5 query points.

The EEG dataset[2] includes raw brainwave signal recordings collected during visual and motor stimulation experiments. The signals were sampled at 128 Hz using electrodes placed at various scalp

---

[1]https://cseweb.ucsd.edu/~yaq007/NASDAQ100_stock_data.html
[2]https://archive.ics.uci.edu/dataset/457/eeg+steady+state+visual+evoked+potential+signals

locations. The dataset contains recordings from a group of subjects participating in different experimental protocols, including SB1, Five Box Visual Test 1; SB2, Five Box Visual Test 2; SB3, Five Box Visual Test 3; SV1, Visual Image Search; and SM1, Motor Images (Hand Shake Experiment). Each protocol involves specific stimuli designed to evoke distinct neural responses, with visual stimulation for the SB and SV tests, and motor imagery for the SM test.

In this study, meta-training episodes are generated from subject `A001SB1_1` (Group A, Subject 001, Test SB1). The first 70% of the time-series data recorded from electrode AF4 for this subject is used to create the meta-training episodes. For the in-range scenario, meta-test episodes are generated from the remaining 30% of the data from the same subject and test. To evaluate the model's performance on out-of-range data, inference episodes are constructed using the time-series data from a different subject, `A003SB1_1`, ensuring clear separation between training, in-range testing, and out-of-range evaluation.

The NASDAQ and EEG datasets provide several advantages over synthetic datasets for model validation, especially in meta-learning. Both datasets feature domain partitioning to validate the generalization performance of algorithms, with distinct training, testing, and out-of-range evaluation episodes that closely mimic real-world scenarios. Our method is implemented using a standard feed-forward neural network to mitigate any bias due to model selection, and this setup also facilitates the visualization of network parametrization with varying inputs, as shown in Figure 4.

For both datasets, we use a three-layer MLP with 64 hidden dimensions and ReLU activation as the base model. The weight and bias distributions are modeled using joint SVGP with 64 inducing points. The training procedure consists of 5,000 iterations with a single batch to ensure the convergence of latent variable parameters. We employ the standard Adam optimizer with a learning rate of 1e-4 within the Bayesian framework. The regularization term consists of the sum of layer-wise KL divergences between the latent variables for the weight and bias terms and their priors. A dual-layer KAN with hidden layer dimension twice the size of input is utilized for IAF modelling.

We provide the visualization of the self-correlation structures of the Gaussian posterior distribution and parameter posterior distribution here for readers to grasp a intuitive understanding of output parameter distribution character.

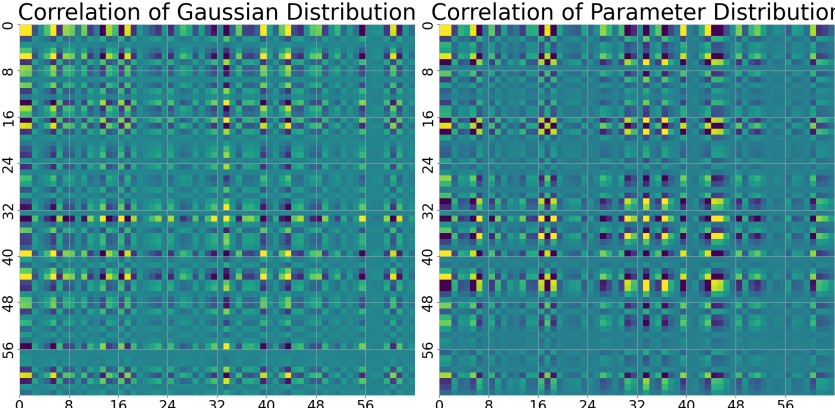

Figure 3: Correlation structure of the Gaussian distribution and parameter distribution approximated by the normalizing flow, calculated on 10 samples across 5 episodes. This visualization confirms that the normalizing flow conversion induces novel correlation structure across parameter components, further refining the precision of posterior distribution representation.

In the visualization result, the off-diagonal bands—particularly in the left-bottom and right-upper regions—reveal an additional correlation structure induced by the IAF layer, which contrasts with the pre-existing diagonal correlation. This observation further demonstrates the effectiveness of transformation of the posterior distribution using a combination of Gaussian Process and a normalizing flow hyper-network.

### A.3.2 OBJECT POSE PREDICTION

Many meta-learning algorithms implicitly assume that datasets are mutually exclusive, meaning that there is no overlap of categories between the training data of one episode and that of another. However, this assumption does not always hold in real-world applications, making standard meta-learning approaches unsuitable for certain domains. To address this challenge, we propose a benchmark for non-mutually exclusive tasks using the Object Pose Prediction dataset[3], which is constructed by rendering the Pascal 3D+ dataset[4] with MuJoCo (Todorov et al., 2012). This dataset consists of grayscale object model images of 10 classes, with each class containing multiple episodes at a resolution of $128 \times 128$. Each episode includes 10 images randomly sampled from 100 renderings of a class. The goal of a meta-learning algorithm is to leverage images across training classes to predict the orientation of an object image relative to a canonical pose, using 5 query samples and 5 support samples.

To solve this task, we adopt a three-layer Convolutional Neural Network (CNN), mimicking the feature extractor used in the original research paper (Yin et al., 2020). Each layer consists of a combination of 2D convolution and ReLU activation. The first two layers are enhanced with HP-GP to enable meta-learning of weights across input classes. The weight and bias distributions are modeled jointly using decoupling SVGP with 128 inducing points. We use an Adaptive Average Pooling operation followed by a convolutional layer with a single-sized kernel to prepare the feature maps for kernel computation, reducing their spatial size and input dimension by a square root factor(with rescaling factor=0.5). The output dimensions of convolutional layers are 32, 64, 64, and 64, and the last two convolutional layers are configured with a stride of 2. Max-pooling layers with stride 2 are placed after all convolutional layers to further reduce the spatial dimensions of the feature maps. A dual-layer KAN with hidden layer dimension identical to input is utilized for IAF modelling.

After the convolutional operations, a three-layer MLP with output sizes 64, 32, and 1 is used to generate the prediction. The training procedure runs for 5,000 iterations and employs the Adam optimizer with a learning rate of 3e-4, optimizing the Bayesian framework. The log-likelihood term in the objective function is represented by the MSE deviation from the predicted target. The regularization term consists of the sum of layer-wise KL divergences between the latent variables for the weight and bias terms and their prior.

A visualization made by T-SNE is provided to illustrate the distribution shift of variables over the calculation pipeline of HP-GP. While the input features exhibit minimal clustering structure, the application of HP-GP—using the weight samples illustrated in the middle row—significantly amplifies the clustering behavior, resulting in clearly distinguishable groups in the output latent space. This enhancement underscores the HP-GP's capacity to extract and emphasize underlying structural patterns, rendering clustered latent feature which is beneficial to the processing pipeline of follow-up layers in the base model.

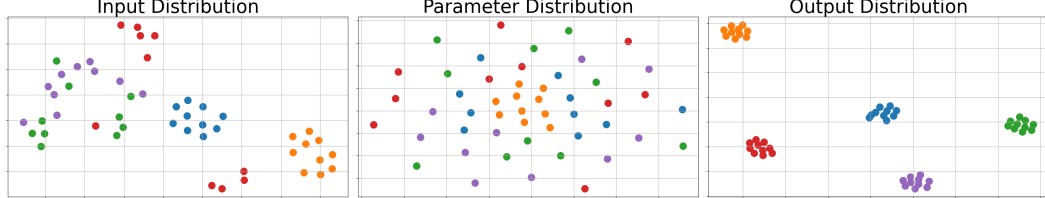

Figure 4: T-SNE visualization of input latent feature, parameter samples, and output latent feature. While input features exhibit minor distribution clustering feature, the clustering is significantly amplified after the processing of HP-GP with weight samples provided in the middle row. Each color represents a distinct episode consisting of 5 support and 5 query samples.

---

[3]https://github.com/google-research/google-research/tree/master/meta_learning_without_memorization

[4]https://cvgl.stanford.edu/projects/pascal3d.html

## A.4 EXTRA EXPERIMENTS

### A.4.1 EFFECT OF HYPER-NETWORK PARAMETRIZATION

Similar to standard deep learning, the parametrization of hyperparameters significantly influences the performance of hyper-networks, affecting training outcomes in meta-learning episodes. In this section, we explore the impact of hyperparameter selection on the performance of HP-GP in the context of meta-learning regression using the NASDAQ dataset. The base network is a three-layer MLP with an output size of $1 \times 64 \times 64 \times 1$ and ReLU activation. The training setup follows the same configuration as described in the Time Series Prediction experiment A.3.1.

We focus on two key elements of hyper-network parameters: (1) Number of IAF layers (2) Kernel function (3) Number of inducing variables. A comprehensive series of experiments is conducted by varying these settings, and the results are presented in Figure 5.

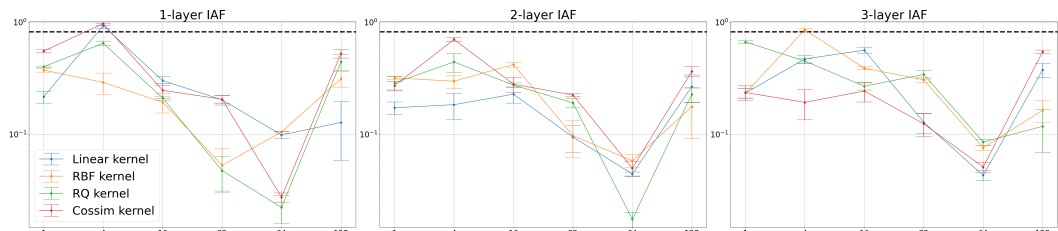

Figure 5: Average Root Mean-Squared Error(RMSE) with standard deviation / 5 comparing different hyper-network parameterizations. The prediction error consistently decreases as the number of inducing variables increases from 4 to 64, but shows a slight increase at 128, suggesting potential over-parameterization. On the other hand, HP-GP delivers the minimal RMSE when combined with two IAF layers, the ascending RMSE in 3-layers IAF also implies the potential over-parameterization of IAF layer composite. The black dashed horizontal line denotes the best performance of a vanilla 3-Layer MLP.

As shown in the experimental outcomes, the performance of HP-GP improves progressively with an increase in training resources, specifically through the increase in the number of inducing variables and IAF layers. In contrast, the performance of HP-GP is relatively unaffected by the choice of kernel function. This is further corroborated by the MSE, which remains consistently lower for HP-GP compared to the vanilla Linear layers, regardless of the specific kernel function used.

### A.4.2 ABLATION ANALYSIS ON FEATURE MAP RESCALING

Since the computation of Gaussian Processes is inherently resource-intensive, it is a natural approach to apply HP-GP only to the layers most crucial for the performance of meta-learning. This raises an important question: How does the adaptation of different layers in the network impact the model's performance? Additionally, the need for adaptation to high-dimensional and varying-scale intermediate feature maps introduces the feature rescaling component in the amortized network. We will investigate the influence of these factors in this section. Specifically, we focus on the impact of layer-wise ablation and the feature and spatial scaling factors.

We focus solely on a single-layer implementation, as we observe that HP-GP performs poorly in meta-testing when applied to multiple layers. Furthermore, the rescaling factor controls how the dimensionality and spatial size of the feature map are reduced.

Each implementation is trained with 5,000 iterations and 32 inducing variables for computational efficiency, with all other training settings consistent with those described in the experimental details section above. The experimental results are presented in Table 3.

According to the experimental results, HP-GP shows a gradual performance improvement with an increasing rescaling factor. In the layer-wise ablation study, the combination of Convolution-1 and Convolution-2 layers yields the most significant improvement, suggesting that implementing hyper-network parametrization on shallower layers is more effective. In contrast, applying HP-GP to the Convolution-3,4 layers results in relatively poor performance across all configurations. This could be partially attributed to numerical instability stemming from resolution scarcity, which is further exacerbated by the use of max pooling rescaling and the minimum 0.3 rescaling factor. When

Table 3: Average Mean-Squared Error (MSE) with standard deviation for HP-GP with different layer applications, Adaptive pooling method and feature rescaling factor. (Over 100 trials.)

| Rescaling factor | Convolution-1 | | Convolution-2 | | Convolution-3 | | Convolution-4 | |
|---|---|---|---|---|---|---|---|---|
| | Average Pooling | Max Pooling | Average Pooling | Max Pooling | Average Pooling | Max Pooling | Average Pooling | Max Pooling |
| 0.3 | $2.2847_{\pm 2.4192}$ | $2.5417_{\pm 2.7629}$ | $2.4089_{\pm 2.3891}$ | $2.4257_{\pm 2.3151}$ | $3.2632_{\pm 2.7577}$ | $3.7315_{\pm 3.1615}$ | $4.0064_{\pm 2.5783}$ | $7.0497_{\pm 5.3465}$ |
| 0.4 | $2.1246_{\pm 2.8752}$ | $2.4273_{\pm 2.9428}$ | $3.1064_{\pm 2.9921}$ | $2.9852_{\pm 2.2637}$ | $2.5841_{\pm 1.3265}$ | $2.8736_{\pm 1.5284}$ | $3.2158_{\pm 1.8426}$ | $4.8732_{\pm 3.2158}$ |
| 0.5 | $2.3471_{\pm 2.3252}$ | $2.2128_{\pm 2.3371}$ | $2.3749_{\pm 2.6769}$ | $2.5194_{\pm 2.7895}$ | $2.5217_{\pm 2.6511}$ | $2.2217_{\pm 2.6512}$ | $2.6472_{\pm 2.8156}$ | $4.3217_{\pm 5.2135}$ |
| 0.6 | $1.6852_{\pm 1.4218}$ | $1.7426_{\pm 1.5632}$ | $1.2563_{\pm 0.7249}$ | $1.3482_{\pm 0.7864}$ | $1.9427_{\pm 1.0425}$ | $2.1564_{\pm 1.1843}$ | $2.8735_{\pm 1.6328}$ | $4.1257_{\pm 2.8735}$ |

both max pooling and average pooling operations are combined, HP-GP's stability shows minimal variation across most parameter combinations. The ablation analysis for HP-GP serves as a reference for model performance when combined with suboptimal feature rescaling implementations, further confirming the performance stability for a robust machine learning model.

### A.4.3 Ablation study on normalizing flow

Although the visualization diagram in Figure 3 establishes a intuitive understanding on the conversion of posterior distribution implemented by the IAF module with KAN implementation, we still need to answer the inquiries into the specific functions of each component within the model. We conduct an additional ablation study on HP-GP, with gradually displacement of normalizing flow components and replacement with MLP transformations. By comparing the eventual outcome further of the comparison, we further demonstrate the functionality of the constituent components of the current HP-GP establishment.

Table 4: Average Mean-Squared Error (MSE) with standard deviation for HP-GP with ablation study on normalizing flow components. (Over 100 trials.)

| Methods | MSE($\downarrow$) | PCC($\uparrow$) | CR($\uparrow$) |
|---|---|---|---|
| W/O IAF | $4.512_{\pm 3.551}$ | $0.598_{\pm 0.462}$ | $0.432_{\pm 0.782}$ |
| MLP replacement | $2.584_{\pm 2.917}$ | $0.738_{\pm 0.298}$ | $0.551_{\pm 0.492}$ |
| IAF w/ MLP | $2.629_{\pm 2.869}$ | $0.752_{\pm 0.286}$ | $0.603_{\pm 0.485}$ |
| IAF w/ Conv1 | $3.756_{\pm 2.771}$ | $0.622_{\pm 0.330}$ | $0.459_{\pm 0.459}$ |
| IAF w/ Attention | $2.470_{\pm 2.206}$ | $0.811_{\pm 0.253}$ | $0.634_{\pm 0.344}$ |
| **HP-GP w/ original IAF** | $\mathbf{2.120}_{\pm 2.845}$ | $\mathbf{0.875}_{\pm 0.236}$ | $\mathbf{0.686}_{\pm 0.310}$ |

As evidenced by the experimental results, the performance of HP-GP exhibits significant degradation with gradual removal of constituent components during the ablation study on the normalizing flow module. The complete removal of IAF (w/o IAF) results in the most severe degradation across all metrics, with significantly elevated MSE ($4.512 \pm 3.551$), reduced PCC ($0.598 \pm 0.462$), and diminished CR ($0.432 \pm 0.782$), indicating substantial impairment in both point prediction accuracy and uncertainty quantification reliability. Interestingly, the two intermediate ablation configurations - IAF replacement with MLP and IAF w/ MLP - exhibit remarkably similar performance with only marginal differences. This contrast may blame on experimental randomness, or suggest potential compensatory mechanisms comparing shallow MLPs and MLPs with sequential input dimension restriction within IAF framework. The preservation of original HP-GP maintains optimal performance across all metrics, validating the potency of shallow KAN components when served as normalizing flow transformation components. The progressive increase in standard deviations across ablation conditions further validates that component removal not only reduces predictive performance but also introduces greater instability in model outputs, highlighting the critical role of normalizing flow components in ensuring both accuracy and reliability of the HP-GP framework.

An additional time complexity analysis is also provided for deeper understanding of the normalizing flow implementation aside from the empirical experimental outcome:

For a MLP layer with $N_{in}$ inputs and $N_{out}$ outputs, the time complexity of its forward propagation is $O(N_{in} \cdot N_{out})$. The computational cost of KAN is dominated by $O(N_{in} \cdot N_{out})$ for each spline evaluation. The standard algorithm for evaluating a $k$-order B-spline is a recursive process with a time complexity of $O(k^2)$, resulting in the overall time complexity of $O(N_{in} \cdot N_{out} \cdot k^2)$.

Based on the calculations, KAN's time complexity differs from MLP's only by a $k^2$ constant term. The theoretical advantage of KAN lies in its superior neural scaling laws, which claim that equivalent accuracy can be achieved with a smaller network width $N$. In this sense, KAN can reduce the computational overhead while maintaining fitting performance.

### A.4.4 Empirical running time comparison

An additional comparison of running time among the three algorithms (NGGP, BMAML and the proposed HP-GP) is provided on the NASDAQ and EEG datasets under in-range scenario. We run each algorithm for 1000 epochs to reduce randomness. The experiments are performed on a single RTX 4090 (24GB) graphical computation card with CUDA Version: 12.6 and PyTorch version: 2.0.1. We record all methods' clock running times in the following table for comparison:

Table 5: Clock running times for algorithms(in seconds) on time series prediction tasks

| Methods | NASDAQ | EEG |
|---------|--------|-----|
| BMAML | 1026.144 | 898.792 |
| NGGP | 15711.125 | 9536.535 |
| **HP-GP** | **558.699** | **420.458** |

As shown in Table 5, due to the efficient hierarchy implementation, our method achieves the lowest clock running time among all methods. This further demonstrates the scalability our proposed method. Note that the running time can be further reduced with more efficient implementations of operators, such as CUDA or Triton.

### A.5 Performance comparisons on Mini-ImageNet

The widely-used Mini-ImageNet dataset(Russakovsky et al., 2015) is comprised of 100 image classes within ImageNet dataset, with 600 images for each class. We follow the standard practice to split the training, validation, and testing sets with 64, 16, and 20 classes, respectively.

To solve this task, we implement HP-GP on a Vision Transformer architecture, adopted from a SoTA modification of the transformer architecture on this dataset (Sun et al., 2022). This model consists of a patch embedding layer to convert input images into sequence of tokens, followed by a series of transformer blocks with multi-head self-attention and MLP layers, and post-processed by a output classifier for dimension conversion into the number of output classes. The patch embedding layer employs the parameterization of 768 embedding dims and 16 patch dimensions. Each transformer block incorporates an attention calculation and a residual connection with MLP conversion and layer normalization. The attention mechanism adopts 8 heads with disabled bias term through QKV calculation. The MLP component within the residual connection is consisting of 3 layers with 768, 3072, 768 dimensions respectively.

Under this framework, HP-GP is applied to the first layer within the MLP component in the residual connection to enable meta-learning of weights across input episodes. The weight and bias distributions are modeled jointly using decoupling SVGP with 768 inducing points. We adopt an Adaptive Average Pooling operation followed by a convolutional layer with a single-sized kernel to prepare the feature maps for kernel computation, reducing their spatial size and input dimension by a square root factor(with rescaling factor=0.5). A dual-layer KAN with hidden layer dimension identical to input is utilized for IAF modelling. Additionally, the kernel function employed in this experiment was the RBF kernel.

The training procedure runs for 10,000 iterations and employs the Adam optimizer with a learning rate of 1e-4, for Bayesian framework optimization. The regularization term consists of the sum of layer-wise KL divergences between the latent variables for the weight and bias terms and their prior.

According to the experimental outcome, the application of HP-GP demonstrates its potential for further enhancement on predictive performance. Throughout our experiment, the direct modification of QKV output with HP-GP demonstrates unsatisfactory performance, which may suggest that a more appropriate joint probability modeling approach needs to be redesigned for outputs with relevance. Nevertheless, applying HP-GP to the MLP or convolutional layers within existing models consis-

Table 6: Comparison of methods on mini-ImageNet dataset

| METHODS | 5-way 1-shot (ACC) | 5-way 5-shot (ACC) |
|---------|--------------------|--------------------|
| METAQDA (Zhang et al., 2021b) | $65.12_{\pm 0.66}$ | $80.98_{\pm 0.75}$ |
| HCTransformer (He et al., 2022) | $74.62_{\pm 0.20}$ | $89.19_{\pm 0.13}$ |
| SSL-ViT-16 (Bhattacharyya et al., 2022) | $86.50_{\pm 0.17}$ | $96.22_{\pm 0.06}$ |
| Vanilla GL-ViT (Sun et al., 2022) | $88.04_{\pm 0.59}$ | $96.45_{\pm 0.20}$ |
| **HP-GP + GL-ViT (RBF)** | $\mathbf{89.25}_{\pm 0.67}$ | $\mathbf{97.01}_{\pm 0.31}$ |

tently yields significant performance improvements in experimental results. This indicates that by modifying or adding relevant structures within the original network, HP-GP can be effectively integrated with various novel deep learning network architectures to achieve further performance gains.

