# OpenReview forum: "Hierarchical Parametrization with Gaussian Process for Bayesian Meta-Learning"
_ICLR.cc/2026/Conference — Submitted to ICLR 2026_

### Official Review · Reviewer_SoTY · 2025-10-29

**Soundness:** 3
**Presentation:** 3
**Contribution:** 3
**Rating:** 6
**Confidence:** 2

**Summary:**

The paper proposes a Bayesian meta-learning method that utilizes Stochastic Variational Gaussian Processes (SVGP) and Normalizing Flows, implementing the amortization network in a layer-wise manner. It shows that SVGP enables input–output tensor decoupling, which reduces computational cost, while Normalizing Flows improve performance by mapping the Gaussian posterior from SVGP to a more expressive distribution.

**Strengths:**

- The idea of using inducing points in SVGP and mapping the posterior to a more expressive, non-Gaussian distribution using Normalizing Flows is innovative. The decoupled calculation derived from the product of two Gaussians also significantly reduces computational complexity.
- It also proposes an additional optimization by leveraging the recently introduced KAN as a substitute for MLPs, which shows a performance improvement in the ablation. An addition that I found adds meaningful value to the paper.
- The paper’s connection to related work is well-researched and presented in a timely and relevant manner.

**Weaknesses:**

- The experimental section could be expanded further. For example, by including the final experiment from [1].
- The limitations of the proposed method should be articulated more clearly in the conclusion section.

[1] Meta-learning without memorization (Yin, et al. 2020)

**Questions:**

__Q1.__ Could you comment on why the performance of HP-GP appears to be relatively unaffected to the choice of kernel function? Additionally, have you considered evaluating a non-smooth kernel, such as the Matérn $\frac{1}{2}$ kernel?

__Q2.__ What is the time comparison between using KAN and other architectures in the Normalizing Flow ablation experiment presented in Section A.4.3?

__Q3.__ Regarding the inducing points $\mathbf{u}$, how is their number determined? Is the number of inducing points learned during training, or must it be specified manually by the user? In any case, it would be helpful if you could provide more details on the choice of number of $\mathbf{u}$ for each experiment. If this is specified manually, an ablation study would add further value to the paper.

__Q4.__ In Table 2, your proposed method uses two different base networks (ResNet-18 and Conv-4). Could you comment on how fair or comparable this setup is relative to the baselines? For example, do the architecture-agnostic baselines also use ResNet-18 or Conv-4? And for the non-agnostic baselines, are they using a comparable number of parameters? As at the moment, the results for the baselines from Yin et al. (2020) appear to be taken directly from their paper.

__Q5.__ I wonder if replacing the Normalizing Flows with Diffusion or Flow Matching models would work in your setting, especially given their strong performance in high-dimensional setups such as neural network weights. Could you comment on this?

__Q6.__ Could you also disclose the complete setup used to produce the t-SNE experiment in Figure 4? It might be helpful to include this information in the Appendix for completeness.

__Minor__

- In line 149, the abbreviation ABML should be expanded. For instance, you could provide the full term Amortized Bayesian Meta Learning (ABML) (Ravi & Beatson, 2019) in line 147.

---

> ### Author Response · Authors · 2025-11-17
> **Rebuttal to Official Review by Reviewer SoTY**
>
> We sincerely thank you for your support of this work. Below, we respond to each of your comments in detail, and we hope these revisions adequately address your concerns.
>
> > **R1: Comment on the role of kernel functions**
>
> Thank you for your positive comments. According to experimental outcome, HP-GP demonstrates sufficient adaptability in kernel selection. We speculate this may stem from two reasons: (1) In the validation experiments, the training data is relatively dense compared to the output function space, diminishing the impact of different kernels. Especially when in the time series experiment, the dimension of input and output space of HP-GP are 1 and 64, supported by 64 inducing variables. (2) The impact of IAF posterior conversion. This structure allows the posterior weight distribution output by HP-GP to extend beyond the initial Gaussian distribution predicted by the kernel function. We can observe that HP-GP's performance is relatively stable against different choice of kernel functions, further demonstrating the effectiveness of the proposed algorithm.
>
> An additional time series prediction experiment with HP-GP+Matern2.5 is provided as follows:
> | METHODS | NASDAQ IN-RANGE(↓) | NASDAQ OUT-OF-RANGE(↓) | EEG IN-RANGE(↓) | EEG OUT-OF-RANGE(↓) |
> |---|---|---|---|---|
> |HP-GP(RQ)|0.066±0.031|0.069±0.035|0.086±0.057|0.088±0.054|
> |HP-GP(Linear)|0.063±0.026|0.075±0.044|0.077±0.049|0.084±0.059|
> |HP-GP(Matern-0.5)|0.072±0.029|0.076±0.040|0.092±0.053|0.095±0.057|
>
> > **R2: Time comparison between KAN and other architectures in Normalizing Flow**
>
> Thank you for your constructive comments.  We have conducted additional comparison among two related algorithms (BMAML, NGGP) on NASDAQ and EEG datasets under OoD scenario. The experiments are performed on a RTX 4090 GPU with CUDA 12.6 and PyTorch 2.0.1. We record the overall clock running times for each algorithm in the following list for comparison:
> |**Clock running time(in seconds)**|NASDAQ|EEG|
> |---|---|---|
> |BMAML|1026.144|898.792|
> |NGGP|15711.125|9536.535|
> |**HP-GP**|**558.699**|**420.458**|
>
> From the results, we can observe that due to the efficient hierarchical implementation, our method surpass previous methods' speed while also demonstrating the best performance.
>
> > **R3: Questions on inducing points**
>
> Thank you for your constructive comments.  Indeed, we already did comparative tests on the number of inducing points in Fig. 5 in Appendix A 4.1. We empirically observed that the number of inducing points equal to output dimension yields optimal performance across different tasks.
>
> > **R4: Response to comparison fairness**
>
> Thank you for your constructive comments.  We selected ResNet-18 and Conv-4 as the foundational architectures for our method implementation, primarily to manifest the memorization problem on object pose prediction dataset. Conv-4  is the standard feature extraction network used in related studies (NGGP and DKT) (Line 417). Therefore, our experiments provide a completely fair comparison against standard GP meta-learning implementations including NGGP and DKT.
>
> > **R5: Response to Flow Matching model adjustment**
>
> Thank you for your insightful comments. Replacing formalized flow with a stream-based modeling approach is an intriguing idea. But to my knowledge, this structure encounters difficulties when integrated into a hypernetwork architecture alongside existing neural networks. In stream-based model predictions, the optimization objective: optimal velocity vector, is obtained through linear interpolation between the initial noise and the predicted result. In meta-learning scenarios, an additional NN must be trained to capture the parameter evolution on each task episode. This approach makes the training cost proportional to the number of training domains included, rendering it impractical for meta-learning tasks involving a large number of diverse task domains. We leave it for future work to explore replacing normalizing flow with diffusion models or flow matching.
>
> > **R6: Clarifications on the t-SNE experiment**
>
> Thank you for your constructive comments.  In this t-SNE visualization experiment, we employ the same configuration as the object pose prediction experiment(Line 1218). This figure visualizes the t-SNE distributions of intermediate variables after resampling adaptive pooling,GP prediction, and final posterior outputs after IAF across input samples. The experimental results demonstrate that the parameter weights and latent features generated by HP-GP exhibit certain degree of clustering across the input samples. We will release all codes for reproducing this paper.

---

> ### Author Response · Authors · 2025-11-19
> **Official Comment by Authors**
>
> (Following the above)
>
> > **R7: Response to expanding experiment scope**
>
> Thank you for the constructive comment.  We further implement HP-GP on a recent SoTA backbone model on the mini-ImageNet classification task and provide the experiment results and comparisons with similar methods for comparison. The result can be found in appendix A4.5 of the rebuttal revision to demonstrate the proposed method's scalability.
>
> |METHODS|5-way1-shot(ACC)↑|5-way5-shot(ACC)↑|
> |---|---|---|
> |METAQDA[1]|65.12±0.66|80.98±0.75|
> |HCTransformer[2]|74.62±0.20|89.19±0.13|
> |SSL-ViT-16[3]|86.50±0.17|96.22±0.06|
> |GL-ViTbaseline[4]|88.04±0.59|96.45±0.20|
> |**HP-GP+GL-ViT(RBF)**|**89.25±0.67**|**97.01±0.31**|
>
> This result further demonstrates the generalization ability of the proposed method to broader scope, which even surpass the strong vision transformer baselines, further demonstrating this method's scalability.
>
>
> > **R8: Discussions on limitations of the proposed method**
>
> We have revised our discussion on the limitation of the proposed method. We believe the two promenint limitation of the proposed method are the study of adaptation to computationally optimized kernels and modern large foundation models. The following works are expected to compensate for the limitations considered.
>
> > **R9: the abbreviation ABML should be expanded**
>
> Thank you for the constructive comment. We have revised our manuscript accordingly.
>
> [1] Zhang, Xueting, et al. "Shallow bayesian meta learning for real-world few-shot recognition." CVPR. 2021.
>
> [2] He, Yangji, et al. "Attribute surrogates learning and spectral tokens pooling in transformers for few-shot learning." CVPR. 2022.
>
> [3] Bhattacharyya, Prarthana, et al. "Visual representation learning with self-supervised attention for low-label high-data regime." arXiv preprint arXiv:2201.08951 (2022).
>
> [4] Sun, Mingze, Weizhi Ma, and Yang Liu. "Global and local feature interaction with vision transformer for few-shot image classification." CIKM 2022.

---

> > ### Comment · Reviewer_SoTY · 2025-11-27
> >
> > Thank you very much for the response.
> >
> > > We have conducted additional comparison among two related algorithms (BMAML, NGGP) on NASDAQ and EEG datasets under OoD scenario.
> >
> > To clarify, my original comment referred to comparing the runtime of KAN with other possible modules in the experiments presented in Appendix A.4.3. However, I see that in line 1401 of the revised manuscript you have included a time-complexity comparison addressing this point. This addition resolves my concern.
> >
> > > Thank you for your constructive comments. We selected ResNet-18 and Conv-4 as the foundational architectures for our method implementation, primarily to manifest the memorization problem on object pose prediction dataset. Conv-4 is the standard feature extraction network used in related studies (NGGP and DKT) (Line 417). Therefore, our experiments provide a completely fair comparison against standard GP meta-learning implementations including NGGP and DKT.
> >
> > After checking the NGGP paper, I could not find any mention of using a Conv-4 architecture. However, it is indeed used in the DKT paper. In addition, NGGP and DKT should be explained more clearly in the manuscript, or at least their full names should be provided when they first appear and add proper citation.
> >
> > Overall writing needs to be improved for the camera ready version if accepted. Given the current state of the manuscript, I will maintain my original score and encourage consideration of the perspectives of the other reviewers.

---

> ### Author Response · Authors · 2025-11-28
>
> Thank you for taking the time to review our rebuttal. Your detailed comments have been indeed helpful in helping us to further improve the quality.
>
> > R1. Comments have been addressed.
>
> We are delighted to know that your comments have been addressed.
>
> > R2. Abbreviations and architecture.
>
> Thank you for your constructive comments. We have further improved the manuscript by adding experiment descriptions with all baseline methods' abbreviations stated clearly (High lighted in blue in Line 364-372 and Line 449 in the current paper). Yes, you are right. DKT used Conv-4 while NGGP used a slightly different architecture (Conv-3). We used the same architecture across all methods to ensure the fairness of comparisons. We have double checked the paper to further improve the writing.
>
> Thank you very much for your supportive review!

---

### Official Review · Reviewer_9yzQ · 2025-11-01

**Soundness:** 2
**Presentation:** 3
**Contribution:** 3
**Rating:** 6
**Confidence:** 4

**Summary:**

The paper proposes HP-GP (Hierarchical Parametrization with Gaussian Process), a Bayesian meta-learning framework that combines Sparse Variational Gaussian Processes (SVGP) and Inverse Autoregressive Flow (IAF) with Kolmogorov–Arnold Networks (KAN) to generate episode-specific neural network weights in a probabilistic, model-agnostic, and scalable manner. HP-GP uses a layer-wise decoupled GP to reduce computational cost and leverages normalizing flows to transform Gaussian posteriors into richer weight distributions. It is evaluated on few-shot regression tasks (time series prediction and object pose estimation), where it consistently outperforms state-of-the-art methods, especially in challenging non-mutually exclusive settings that induce memorization.

**Strengths:**

The main strength of the proposed approach are:

* Model-agnostic and lightweight. The method requires minimal architectural changes required, and can be integrated into existing networks.

*Strong empirical performance. The proposed approach achieves SOTA results on both in-distribution and out-of-distribution few-shot regression benchmarks.

*Robust to memorization. The approach excels in non-mutually exclusive tasks by leveraging cross-class information via inducing variables.

*Theoretically grounded. The method is theoretically grounded and provides bounds on norm, Rademacher complexity, and approximation error using RKHS and Barron space frameworks.

* Uncertainty-aware. Finally, being rooted on Gaussian Processes, the method naturally supports uncertainty quantification through Bayesian inference.

**Weaknesses:**

The main limitations of the paper are:

* Computational overhead: Despite decoupling, SVGP + IAF introduces nontrivial compute/memory costs vs. simpler meta-learners (e.g., MAML). It would also be useful to compare the method against existing probabilistic methods such as BMAML (Yoon et al.) to further elucidate its strengths/weaknesses when it comes to computational usage.

* Limited task scope. The proposed method is evaluated only on regression (not classification) tasks, whereas existing methods such as BMAML, which is also probabilistic, is evaluated on classification, active learning, and reinforcement learning tasks. I believe that this reduces the generalizability claims of the paper.

* Hyper-parameter sensitivity. The performance of the proposed approach hinges heavily on kernel choice, number of inducing points, and IAF depth (risk of over-parameterization).

* Scalability concerns. While layer-wise, the extension of the paper to very deep architectures (e.g., transformers) is not demonstrated.

* Ablation gaps. Because of the limited analysis of the relative contribution of KAN vs. standard MLPs beyond one table, it is unclear to me if KAN is essential.

**Questions:**

Please refer to the section above.

---

> ### Author Response · Authors · 2025-11-17
> **Rebuttal to Official Review by Reviewer 9yzQ**
>
> We sincerely thank you for your support of this work. Below, we respond to each of your comments in detail, and we hope these revisions adequately address your concerns.
>
> > **R1: Response to computational overhead**
>
> Thank you for your positive and constructive comments. We have conducted additional comparison among 3 related algorithms (BMAML, NGGP, DKT) on NASDAQ and EEG datasets under OoD scenario. We illustrate the decline in MSE as the number of training epochs increases in appendix A4.4 of the rebuttal revision. The experiments are performed on a RTX 4090 GPU with CUDA 12.6 and PyTorch 2.0.1. We take record of the overall empirical running times for each algorithm in the following list for comparison:
> |**Clock running time(in seconds)**|NASDAQ|EEG|
> |---|---|---|
> |BMAML|1026.144|898.792|
> |NGGP|15711.125|9536.535|
> |**HP-GP**|**558.699**|**420.458**|
>
> > **R2: Response to Limited task scope**
>
> Thank you for your constructive comments. We have added an mini-ImageNet experiments with vision transformer (ViT) backbone to further demonstrate the proposed method's generalizability (See R4). We leave it for future work to extend this method for other tasks, such as active learning and reinforcement learning.
>
> > **R3: Response to Hyper-parameter sensitivity**
>
> Thank you for your constructive comment. To fully investigate the hyperparameter sensitivity, we have conducted comparative experiments, including investigations in subsection A.4.1 on kernel functions, number of inducing points, and number of configuration layers; comparative studies in subsection A.4.2 on scaling parameters, pooling methods, and number of configuration layers; and ablation research in subsection A.4.3 on regularization flow. The impact of hyperparameters adjustment confirms the functionality of these components within the overall framework. However, the impact of each hyper-parameter is negligible compared to the performance differences with other methods, strongly indicating the overall stability of HP-GP.
>
> > **R4: Response to Scalability concerns**
>
> Thank you for the constructive comment. We further implement HP-GP on a recent SoTA backbone model on the mini-ImageNet dataset in appendix A4.5 of the rebuttal revision to demonstrate the proposed method's scalability.
>
> |METHODS|5-way1-shot(ACC)↑|5-way5-shot(ACC)↑|
> |---|---|---|
> |METAQDA[1]|65.12±0.66|80.98±0.75|
> |HCTransformer[2]|74.62±0.20|89.19±0.13|
> |SSL-ViT-16[3]|86.50±0.17|96.22±0.06|
> |GL-ViTbaseline[4]|88.04±0.59|96.45±0.20|
> |**HP-GP+GL-ViT(RBF)**|**89.25±0.67**|**97.01±0.31**|
>
> From the results above, we can observe that our proposed method HP-GP achieves the best result among the strong large-scale transformer-based baselines. This demonstrate that the proposed HP-GP can can scale to larger models, such as the vision transformers, on the standard mini-ImageNet dataset. We believe this result further justifies the scalability of the HP-GP. The additional experiment results are added in the Appendix A.5 in the revised manuscript.
>
> > **R5: Response to ablation gaps**
>
> In our approach, KAN serves as a specific implementation of IAF in replacement of MLP, aiming to leverage the network architecture to achieve superior fitting performance with minimal computation overhead. An empirical effectiveness demonstration of the combanation of decopuling kernel realization and the KAN-based IAF calculation can be found in appendix A4.4 in rebuttal revision. The time complexity analysis can be conducted as follows:
>
> For a MLP layer with $N_{in}$ inputs and $N_{out}$ outputs, the time complexity of its forward propagation is $O(N_{in} \cdot N_{out})$. The computational cost of KAN is dominated by $O(N_{in} \cdot N_{out})$ for each spline evaluation. The standard algorithm for evaluating a $k$-order B-spline is a recursive process with a time complexity of $O(k^2)$, resulting in the overall time complexity of $O(N_{in} \cdot N_{out}\cdot k^2)$.
>
> Based on the calculations, KAN's time complexity differs from MLP's only by a k^2 constant term.  While as shown in the original paper, the theoretical advantage of KAN lies in its superior neural scaling laws. Thus, we adopt KAN in this paper to reduce the computational overhead while maintaining fitting performance.
>
> An additional comparison experiment with other common network implementations of IAF is provided as follows:
>
> |METHODS|MSE(↓)|PCC(↑)|CR(↑)|
> |---|---|---|---|
> |w/o IAF|4.512±3.551|0.598±0.462|0.432±0.782|
> |MLP REPLACEMENT|2.584±2.917|0.738±0.298|0.551±0.492|
> |IAF w/ MLP|2.629±2.869|0.752±0.286|0.603±0.485|
> |IAF w/ CONV1|3.756±2.771|0.622±0.330|0.459±0.459|
> |IAF w/ ATTENTION|2.470±2.206|0.811±0.253|0.634±0.344|
> |**HP-GP w/Orignal IAF**|**2.120±2.845**|**0.875±0.236**|**0.686±0.310**|

---

> ### Author Response · Authors · 2025-11-19
>
> (Following the above) In the table provided above, the experimental outcome of IAF implementation with Conv1 and attention is added for comparison. According to the ablation result, the original combination of implementation(HP-GP with KAN) delivers the best predictive performance, validating the necessity of implementation based on KAN structure.
>
> [1] Zhang, Xueting, et al. "Shallow bayesian meta learning for real-world few-shot recognition." ICCV 2021.
>
> [2] He, Yangji, et al. "Attribute surrogates learning and spectral tokens pooling in transformers for few-shot learning." CVPR 2022.
>
> [3] Bhattacharyya, Prarthana, et al. "Visual representation learning with self-supervised attention for low-label high-data regime." arXiv preprint arXiv:2201.08951 (2022).
>
> [4] Sun, Mingze, Weizhi Ma, and Yang Liu. "Global and local feature interaction with vision transformer for few-shot image classification." CIKM 2022.

---

> ### Comment · Reviewer_9yzQ · 2025-11-25
>
> Glad to see the additional experiments. I am satisfied with the results.

---

> > ### Author Response · Authors · 2025-11-26
> >
> > Thank you for taking the time to review our rebuttal and for reconsidering your evaluation. Your insightful comments have been extremely helpful in improving the quality the paper. We sincerely appreciate your review.

---

### Official Review · Reviewer_sjk2 · 2025-11-02

**Soundness:** 3
**Presentation:** 2
**Contribution:** 2
**Rating:** 4
**Confidence:** 3

**Summary:**

The paper proposes a new meta-learning method that integrates Gaussian Process into the modeling. Conventionally, probabilistic meta-learning often relies on a hyper-model (also known as hyper-net) to generate task-specific parameters to adapt to the task of interest. Instead of following a parametric approach, the paper proposes to use Gaussian Process coupled with normalizing flow. Such a modeling is claimed to be flexible and scalable with minimal change to the model used. Empirical evaluation shows that the newly-proposed method outperforms existing meta-learning methods on some benchmarks.

**Strengths:**

The paper provides an extensive study about related studies and presents detailed background knowledge (especially section 3). These make the paper self-contained and easier to follow.

The paper also includes theoretical analysis on the generalization of the proposed methods, and in particular upper bounding the error in the form of PAC learning. This provides certain level of generalization guarantee.

The proposed method is evaluated on two families of tasks: one is about time series prediction and the other about object pose prediction. The proposed method has been demonstrated to out-perform existing meta-learning methods in both evaluation benchmarks.

**Weaknesses:**

In the current form, the paper is too dense. The notations used are not well explained. For example: at line 203, "u and $\omega$ are inducing latent functions and design functions". It is unclear what these functions are. In addition, the generative process in the modeling shown in Fig. 1 is not explained at all. Could the authors provide the data generation process like other graphical model papers do? For example, the authors could refer to the template in section 3 of the paper: "Latent Dirichlet Allocation". That would help to understand how data is generated, what the role of each parameter and which kind of parameter inference approach the paper is proposing. In addition, introducing some variables like the description right after Eq. (3) is not recommendable because it suddenly jumps to a much later equation. This causes surprise and hard to understand the paper.

The usage of normalizing flow introduced in the paper is not clear. Could the authors elaborate why normalizing flow is needed, while other hyper-net based meta-learning methods do not?

The empirical evaluation is not too standard to the meta-learning literature. I am aware of the standard evaluation may be old. However, it would be fairer to evaluate on the standard few-shot learning on mini-ImageNet, then move to the new datasets like the one considered in the paper. In addition, the current baselines are outdated (most baselines were published before Covid-19). Should newer baselines be included to reflect the high-performance of the proposed method.

**Questions:**

Please refer to the weaknesses.

---

> ### Author Response · Authors · 2025-11-17
> **Rebuttal to Official Review by Reviewer sjk2**
>
> We sincerely thank you for your thoughtful and detailed feedback, which greatly helps us to further improve this paper. Below, we respond to each of your comments in detail, and we hope these revisions adequately address your concerns.
>
> > **R1: Explanation on notation confusion**
>
> Thank you for your constructive comments. Very sorry for the confusion. We make clarifications as follows: in this sentence, we introduce the definition of the Sparse Variational Gaussian Prior (SVGP) [10].  In SVGP, $u$ and $\omega$ are inducing function values and design function values, respectively. $z$ and $x$ are the corresponding inducing locations and design locations. We modify the manuscript (line 212-215), which is highlighted in blue for clarity.
>
> >  **R2: Explanation on confusion on Fig. 1**
>
> Thank you for the comment. We add the following additional explanations about Fig. 1 in Section 4.1, highlighted in blue in the revised manuscript for clarification.
>
> -----
>
> Given each input-output pair $(x_{t, i}, y_{t, i})$ from $t$-th task episode, the learning objective is to train a meta-Bayesian model comprising the induced variables $z, u$ for the Gaussian posterior $\omega_t$ (Equation 3 in the paper). Then, with the Gaussian posterior $\omega_t$, we induce the IAF parameters $w$ for the deep network's posterior $\theta_t$ (Equation 4 in the paper)), which is leveraged for final prediction. This thereby enables the model to approximate the complex posterior distribution for meta-learning. For meta-training scheme, the training process is repeated for each task episode $\mathcal{D}_t$. For meta-testing scheme, given support set and query set, HP-GP generates the Gaussian posterior $\hat{\omega}$, and then the deep network's posterior $\hat{\theta}$ for prediction.
>
> ------
>
> > **R3: Introduction of variables like the description right after Eq. (3).**
>
> Thank you for your constructive comments.  We have reorganized the texts in the manuscript. To efficiently implement Eq. (3), we introduce a decoupling realization, which will be elaborated later in Section 4.2. This may help readers the focus on the whole picture and dive into details gradually.
>
> > **R4: Elaborations on the necessity of normalizing flow in HP-GP**
>
> In our research, the introduction of a normalizing flow structure stems from the inadequacy of a purely Gaussian distribution as the posterior modeling approach for network parameters. Relevant studies [1,2,3] indicate that Bayesian neural networks exhibit questionable performance when relying on Gaussian posterior modeling, necessitating empirical adjustments such as cold-starting to enhance practical network performance. Therefore, we opt to incorporate IAF as a posterior correction model. It should be noted that this approach has been preliminarily demonstrated in previous studies [4,5]. However, those studies did not extend to the extent of designing a forward-propagating hyper-network into existing neural networks, which is ultimately fulfilled in our research. Empirical results on various datasets demonstrates the effectiveness of this module in the proposed method.
>
> > **R5: More Empirical evaluations on the mini-ImageNet**
>
> Thank you for the constructive comment. We further implement HP-GP on a recent SoTA backbone model GL-ViT on the mini-ImageNet dataset and provide the experiment results and comparisons with similar methods in the following table:
>
> |METHODS|5-way1-shot(ACC)↑|5-way5-shot(ACC)↑|
> |---|---|---|
> |METAQDA[9]|65.12±0.66|80.98±0.75|
> |HCTransformer[8]|74.62±0.20|89.19±0.13|
> |SSL-ViT-16[7]|86.50±0.17|96.22±0.06|
> |GL-ViTbaseline[6]|88.04±0.59|96.45±0.20|
> |**HP-GP+GL-ViT(RBF)**|**89.25±0.67**|**97.01±0.31**|
>
> From the results above, we can observe that our proposed method HP-GP achieves the best result among the strong large-scale transformer-based baselines. This demonstrate that the proposed HP-GP can can scale to larger models, such as the vision transformers, on the standard mini-ImageNet dataset. We believe this result further justifies the scalability of the HP-GP. The additional experiment results are added in the Appendix A.5 in the revised manuscript.

---

> ### Author Response · Authors · 2025-11-19
>
> [1] Fortuin, Vincent, et al. "Bayesian neural network priors revisited." arXiv preprint arXiv:2102.06571 (2021).
>
> [2] Wenzel, Florian, et al. "How good is the bayes posterior in deep neural networks really?." arXiv preprint arXiv:2002.02405 (2020).
>
> [3] Tran, Ba-Hien, et al. "All you need is a good functional prior for Bayesian deep learning." Journal of Machine Learning Research 23.74 (2022): 1-56.
>
> [4] Sendera, Marcin, et al. "Non-gaussian gaussian processes for few-shot regression." Advances in Neural Information Processing Systems 34 (2021): 10285-10298.
>
> [5] Maroñas, Juan, et al. "Transforming Gaussian processes with normalizing flows." International Conference on Artificial Intelligence and Statistics. PMLR, 2021.
>
> [6] Sun, Mingze, Weizhi Ma, and Yang Liu. "Global and local feature interaction with vision transformer for few-shot image classification." CIKM. 2022.
>
> [7] Bhattacharyya, Prarthana, et al. "Visual representation learning with self-supervised attention for low-label high-data regime." arXiv preprint arXiv:2201.08951 (2022).
>
> [8] He, Yangji, et al. "Attribute surrogates learning and spectral tokens pooling in transformers for few-shot learning." CVPR, 2022.
>
> [9] Zhang, Xueting, et al. "Shallow bayesian meta learning for real-world few-shot recognition." CVPR, 2021.
>
> [10] Hugh Salimbeni, et al.  "Doubly Stochastic Variational Inference for Deep Gaussian Processes."  NeurIPS, 2017

---

### Author Response · Authors · 2025-11-20
**Overall response**

We thank all reviewers for their careful reading, constructive feedback, and positive recognition of our paper’s contributions. We are encouraged that reviewers consistently highlighted the novelty, strong empirical performance, theoretical grounding, and model-agnostic scalability of our proposed HP-GP framework. Below we summarize the core strengths acknowledged by the reviewers and provide a consolidated overview of how we have addressed the major questions and concerns.

***Strengths highlighted by reviewers***

- **Novel Bayesian meta-learning framework.** Reviewers emphasized the novelty of leveraging the decoupled modeling within the stochastic variational Gaussian process (SVGP) framework and normalizing flows to produce expressive, uncertainty-aware posteriors.

- **Model-agnostic and lightweight.** The proposed method can be integrated seamlessly with existing architectures, requiring minimal changes, and supports both classic convolutional and transformer-based backbones.

- **Strong empirical results with theoretical justification** Reviewers noted the consistent SOTA performance across in-distribution, OOD, and non-mutually-exclusive task settings, demonstrating robustness against memorization.

***Summary of key questions and answers***

- **Clarification of notation, model generative process**  We refined the notation and provided clear explanations for the deduction of the proposed algorithmic framework. A detailed step-by-step algorithmic description for HP-GP has been added in Section 4.1 , complementing Fig 1.

- **Experimental scope and additional baselines** We added new classification experiments on mini-ImageNet using a modern ViT backbone (GL-ViT), demonstrating that HP-GP scales to large models and remains SOTA among strong transformer-based competitors. We included runtime comparisons and computational analyses vs. BMAML and NGGP—showing that HP-GP is significantly faster or competitive with baselines despite being more expressive.

- **Ablations, hyperparameter sensitivity**
   We have performed extensive new ablations on:
   -  kernel functions (including Matérn 2.5),
   -  number of inducing points,
   -  IAF depth, pooling strategies, and scaling parameters,
   -  alternative IAF architectures (MLP, conv, attention).

   These studies demonstrate the proposed method is not sensitive to hyperparameter selections and justify the use of KAN-based IAF, which offers better predictive performance with minimal computational overhead.

We appreciate the reviewers’ thoughtful assessments, which helped us substantially improve the clarity, completeness, and empirical breadth of the paper. We hope the revisions address all comments. Should there be any remaining questions or uncertainties, we would be pleased to provide further clarification.

---

### Author Response · Authors · 2025-12-01
**Summary of discussion for AC**

We sincerely thank AC and the Program Committee for the tremendous efforts in sustaining the integrity of the ICLR review process due to the recent Openreview issue.  We are encouraged that reviewers consistently highlighted the novelty, strong empirical performance, theoretical grounding, and model-agnostic scalability of our proposed HP-GP framework. Below, we summarize the core strengths acknowledged by the reviewers and then provide an overview of comments and responses during the discussion period.

***Strengths highlighted by reviewers***

- **Novel Bayesian meta-learning framework.** Reviewers emphasized the novelty of leveraging the decoupled modeling within the stochastic variational Gaussian process (SVGP) framework and normalizing flows to produce expressive, uncertainty-aware posteriors.

- **Model-agnostic and lightweight.** The proposed method can be integrated seamlessly with existing architectures, requiring minimal changes, and supports both classic convolutional and transformer-based backbones.

- **Strong empirical results with theoretical justification** Reviewers noted the consistent SOTA performance across in-distribution, OOD, and non-mutually-exclusive task settings, demonstrating robustness against memorization.

***Discussions summary***

 - **1.Reviewer 9yzQ**
    - Initial comment: Computational overhead, expanding task scope, hyper-parameter sensitivity, scalability to modern vision transformers, ablations
    - Author response: Add clock-time running experiments, mini-Imagenet experiments with vision transformer backbone, hyper-parameter sensitivity experiments, and more ablation results.
    - Further response: ``Glad to see the additional experiments. I am satisfied with the results.’' Raise the score to 8.

 - **2.Reviewer SoTY**
   - Initial comment: Module variants, computational time, more baselines.
   - Author response: Add module variants experiments. Compare computational with baselines. Extending to more baseline comparisons.
   - Further response:  ``This addition resolves my concern.’' Keep the score 6. Further asked for adding full names for baselines. We have added the requested descriptions (highlighted in blue in Line 364-373) and have been waiting for further response.

 - **3.Reviewer sjk2**
   - Initial comment: Notation clarification, additional empirical experiments.
   - Author response: Add notation clarification point-by-point, add additional empirical experiments.  Each modification is highlighted in blue in the modified paper.
   - Further response: The reviewer has not yet replied to our response but we are confident that we have addressed all the comments and improved the paper accordingly.

***Our final scores during the discussion period are 8, 6, 4***

***REBUTTAL OUTCOME HIGHLIGHT***

Reviewers generally indicated that our rebuttal effectively addressed their core comments. Reviewer 9yzQ is glad to see the additional experiments and is satisfied with the results. Reviewer SoTY confirms that the concerns was solved. Reviewer sjk2 does not yet have time to reply but we are confident that we have addressed all the comments. Overall, the discussion showed a clear convergence: the empirical evaluation was more sufficient, the notations were clarified.

We thank AC and all reviewers again for their thoughtful reading, constructive feedback, and recognition of our paper’s contributions.

---

### Meta-Review · Area_Chair_rM54 · 2026-01-01

**Summary:**

This paper proposes a method called Hierarchical Parameterization with Gaussian Process (HP-GP). This is a novel probabilistic meta-learning method that leverages the power of Gaussian Process. By implementing the amortization network layer-wise with decoupling variational Gaussian Process and normalizing flow, HP-GP offers probabilistic parameterization for meta-learning while requiring minimal modifications to the network architecture. This enables flexible and scalable integration of meta-learning into existing neural networks. Our experiments demonstrate the flexibility and robust generalization of HP-GP, outperforming other popular meta-learning methods. The reviewers mention that this is an original work with strong theoretical grounding, and state-of-the-art empirical performance across few-shot regression tasks, including scalability to vision transformers. However, they also highlight some limitations of the paper. In particular, they indicated that in the current form, the paper is too dense. The notations used are not well explained. This was partially addressed in the rebuttal. The reviewers also pointed out that the usage of normalizing flows introduced in the paper is not clear. This was partially addressed in the rebuttal. The reviewers also indicated that the empirical evaluation is not too standard to the meta-learning literature. Other limitations include the computational overhead, SVGP + IAF introduce nontrivial costs vs. simpler meta-learners. Another weakness is that the proposed method is evaluated only on regression (although the rebuttal added some classification experiments). Moreover, the reviewers have indicated that the performance of the proposed approach hinges heavily on kernel choice. Some reviewers also asked for ablation studies which were carried out partially in the rebuttal. Summing up, the paper had concerns around clarity, computational overhead, and practical applicability. The initial submission was hard to follow, with dense notation and limited experimental scope, and although rebuttal improvements added classification results and ablations, making the paper better, I still believe that all this makes the paper close to be borderline. My overall feeling is that most concerns were only addressed partially with small experiments carried out during the rebuttal.

**Reviewer Concerns:**

The rebuttal addressed several major reviewer concerns. It clarified notation and justified the use of normalizing flows, and added extra experiments including mini-ImageNet classification. Authors also provided runtime comparisons showing competitive efficiency. They also conducted ablations studies. Additional clarifications covered hyper-parameter sensitivity, and t-SNE setup. Newer baselines and fair architecture comparisons were also included in some experiments. However, some issues remain partially unresolved: the manuscript still needs improved writing for clarity, and there is some need of showing broader applicability. I also believe that some concerns related to computational overhead versus simpler methods persist. Scalability to very large foundation models and exploration of alternatives like diffusion or flow matching were left for future work.

**Reviewer Scores:**

Only reviewer 9yzQ acknowledges the results provided in the rebuttal and could have increased the given score. Reviewer sjk2 may have also increased a bit the given score since extra experiments were carried out.

---

### Decision · Program_Chairs · 2026-01-26

Reject